# The chromatin remodeling protein CHD-1 and the EFL-1/DPL-1 transcription factor cooperatively down regulate CDK-2 to control SAS-6 levels and centriole number

Jyoti Iyer[1☯]*, Lindsey K. Gentry[2☯], Mary Bergwell[1], Amy Smith[1], Sarah Guagliardo[2], Peter A. Kropp[2], Prabhu Sankaralingam[2], Yan Liu[2], Eric Spooner[3], Bruce Bowerman[4], Kevin F. O'Connell[2]*

1 Department of Chemistry and Biochemistry, University of Tulsa, Tulsa, Oklahoma, United States of America, 2 Laboratory of Biochemistry and Genetics, National Institutes of Diabetes and Digestive and Kidney Diseases, NIH, Bethesda, Maryland, United States of America, 3 Proteomics Core Facility, Whitehead Institute for Biomedical Research, Cambridge Massachusetts, United States of America, 4 Institute of Molecular Biology, University of Oregon, Eugene, Oregon, United States of America

☯ These authors contributed equally to this work.
* jgi2708@utulsa.edu (JI); Kevino@nih.gov (KFO)

**Data Availability Statement:** Our RNAseq data has been added to the Sequence Read Archive and is accessible via this link: https://www.ncbi.nlm.nih.

## Abstract

Centrioles are submicron-scale, barrel-shaped organelles typically found in pairs, and play important roles in ciliogenesis and bipolar spindle assembly. In general, successful execution of centriole-dependent processes is highly reliant on the ability of the cell to stringently control centriole number. This in turn is mainly achieved through the precise duplication of centrioles during each S phase. Aberrations in centriole duplication disrupt spindle assembly and cilia-based signaling and have been linked to cancer, primary microcephaly and a variety of growth disorders. Studies aimed at understanding how centriole duplication is controlled have mainly focused on the post-translational regulation of two key components of this pathway: the master regulatory kinase ZYG-1/Plk4 and the scaffold component SAS-6. In contrast, how transcriptional control mechanisms might contribute to this process have not been well explored. Here we show that the chromatin remodeling protein CHD-1 contributes to the regulation of centriole duplication in the *C. elegans* embryo. Specifically, we find that loss of CHD-1 or inactivation of its ATPase activity can restore embryonic viability and centriole duplication to a strain expressing insufficient ZYG-1 activity. Interestingly, loss of CHD-1 is associated with increases in the levels of two ZYG-1-binding partners: SPD-2, the centriole receptor for ZYG-1 and SAS-6. Finally, we explore transcriptional regulatory networks governing centriole duplication and find that CHD-1 and a second transcription factor, EFL-1/DPL-1 cooperate to down regulate expression of CDK-2, which in turn promotes SAS-6 protein levels. Disruption of this regulatory network results in the overexpression of SAS-6 and the production of extra centrioles.

gov/sra/PRJNA814252. The accession number is PRJNA814252.

**Funding:** KFO is an intramural researcher with the NIH and is directly funded by the NIH. This research was supported by the Intramural Research Program of the NIH, The National Institute of Diabetes and Digestive and Kidney Diseases (NIDDK). The NIH/NIGMS COBRE Grant 5P20GM103636 provided salary support to JI. The funders had no role in study design, data collection and analysis, decision to publish, or preparation of the manuscript.

**Competing interests:** The authors have declared that no competing interests exist.

## Author summary

Centrioles are cellular constituents that play an important role in cell reproduction, signaling and movement. To properly function, centrioles must be present in the cell at precise numbers. Errors in maintaining centriole number result in cell division defects and diseases such as cancer and microcephaly. How the cell maintains proper centriole copy number is not entirely understood. Here we show that two transcription factors, EFL-1/DPL-1 and CHD-1 cooperate to reduce expression of CDK-2, a master regulator of the cell cycle. We find that CDK-2 in turn promotes expression of SAS-6, a major building block of centrioles. When EFL-1/DPL-1 and CHD-1 are inhibited, CDK-2 is overexpressed. This leads to increased levels of SAS-6 and excess centrioles. Our work thus demonstrates a novel mechanism for controlling centriole number and is thus relevant to those human diseases caused by defects in centriole copy number control.

## Introduction

Centrioles are small cylindrical-shaped protein complexes that function in a variety of important cellular processes [1]. In mitotic cells, a centriole pair recruits a proteinaceous matrix of pericentriolar material, or PCM, to form the primary microtubule-organizing center (MTOC) known as a centrosome. Centrosomes, which possess the ability to nucleate and anchor microtubules, can control cell polarity, organize the poles of the mitotic spindle, and specify the orientation of cell division. In nonmitotic cells, centrioles can serve as basal bodies to organize cilia and flagella, and thus contribute to cell motility and signaling.

One key property of centrioles that is critically important for their proper function, is numerical control [2]. That is, centrioles must be present in cells at precisely defined numbers; typically, one or two pairs depending on the cell cycle stage. Not surprisingly, defects in centriole number and structure have been linked to a growing number of human diseases. It has long been known that many cancers are associated with an excess number of centrioles, and it has recently been shown in a mouse model that an experimental increase in centriole number can induce tumor formation in many different tissues [3]. Other diseases linked to centriole dysfunction include primordial dwarfism and primary microcephaly [4–6]. The hallmark of primary microcephaly is a small brain size caused by a defect in expansion of neural progenitor cells in the developing neocortex. Interestingly, while microcephaly can arise as a consequence of an insufficient number of centrioles, it can also arise due to the overproduction of centrioles [7,8], thereby illustrating the critical importance of maintaining proper centriole number during brain development.

Numerical control of centrioles relies upon a single precise duplication event that takes place during each S phase wherein a single daughter centriole is assembled at a perpendicular angle next to each pre-existing mother centriole. The core centriole assembly pathway is broadly conserved and was first elucidated in the nematode *C. elegans*. The master regulator of centriole duplication is a polo-like kinase referred to as ZYG-1 in *C. elegans*, or Plk4 in vertebrates and flies [9,10]. A key event in centriole assembly is the recruitment of ZYG-1 to the site of centriole assembly through a physical interaction with its centriole receptor SPD-2 [11]. ZYG-1 in turn recruits a complex of two proteins, SAS-5 and SAS-6 [12,13] that form the central scaffold that establishes the nine-fold radial symmetry of the centriole [14,15]. This step involves a direct physical interaction between ZYG-1 and SAS-6 [16]. Finally, the microtubule-binding protein SAS-4 is added to the outer wall where it promotes the assembly of the centriolar microtubules [12,13].

How centriole assembly is restricted to avoid an excess number of centrioles has been an active area of research. Numerous studies have demonstrated the need to control the abundance of centriole assembly factors, as experimental overexpression of ZYG-1/Plk4, SAS-5/STIL or SAS-6 results in the formation of extra centrioles [17–23]. To date, most of the mechanisms that have been described function post-translationally to limit the levels of these factors via ubiquitin-mediated degradation (reviewed in [24]).

However, we and others have also demonstrated that the abundance of centriole assembly factors is regulated at a transcriptional level. In particular, members of the E2F family of transcription factors play both positive and negative roles in regulating centriole assembly in vertebrates and invertebrates [25–29]. The E2F transcription factors are obligate heterodimers containing one E2F protein and one DP protein (EFL and DPL respectively in *C. elegans*) [30]. In *C. elegans*, EFL-1 is most closely related to vertebrate E2F4 and E2F5, which primarily act to repress transcription of target genes. Together with DPL-1, the sole *C. elegans* DP protein, EFL-1 negatively regulates centriole assembly, as loss of this transcription factor results in elevation of SAS-6 protein levels and suppression of the centriole assembly defect in animals compromised for ZYG-1 function [27]. EFL-1/DPL-1, however, does not act directly by negatively regulating transcription of the *sas-6* gene [27] and thus how its loss results in elevation of SAS-6 protein levels remains an open question. While E2F-dependent pathways appear to predominate in the transcriptional control of centriole assembly, E2F-independent pathways are likely to exist.

CHD1 (chromodomain helicase DNA-binding protein 1) is a highly conserved chromatin remodeler that arranges nucleosomes at regular intervals across gene bodies in an ATP-dependent manner [31,32]. CHD1 has chiefly been implicated in regulating transcriptional elongation [33–38] but evidence has emerged suggesting that it also plays a role in transcriptional initiation [39–41] and termination [42–44]. Outside of its known roles in regulating gene expression, CHD1 has also been shown to function in DNA repair [45], sister chromatid cohesion [46], and histone modification [35,47]. CHD1 possesses two tandem chromodomains at its N-terminus that selectively bind lysine 4-methylated histone H3 of nucleosomes within transcriptionally-active chromatin [47,48,49], a central SNF2-like ATPase/helicase domain and a C-terminal SANT/SLIDE DNA-binding domain. In humans, CHD1 functions as a tumor suppressor and is the second most frequently deleted gene in prostate cancers [50].

Several lines of evidence suggest that CHD1 might be involved in regulating centriole duplication and/or function. Firstly, CHD1 localizes to the promoters of centrosome genes in mouse and human embryonic stem cells [51,52]. Secondly, CHD1 physically associates with the *Xenopus* ZYG-1 ortholog [53], and finally, *chd1-null* flies have spindle and astral microtubule defects [54]. Here we investigate a role for the *C. elegans* CHD1 protein (CHD-1) in regulating centriole duplication and find that complete loss of CHD-1 or inactivation of its ATPase activity, partially suppresses both the embryonic lethality and centriole duplication defects caused by a hypomorphic *zyg-1* mutation. Loss of CHD-1 is also associated with an increase in the levels of the ZYG-1-binding proteins SPD-2 and SAS-6, likely providing a mechanism for suppression. We present evidence that CHD-1 does not regulate transcription of the *sas-6* gene, but rather negatively regulates CDK-2, which in turn promotes SAS-6 protein levels. We also find that EFL-1/DPL-1 acts in parallel to CHD-1 to downregulate CDK-2, and that the combined loss of both factors leads to further elevation of SAS-6 levels and the appearance of extra centrosomes and multipolar spindles. Our results thus reveal how two transcriptional regulatory pathways are integrated with a CDK-2-dependent mechanism that post-translationally controls SAS-6 expression and centriole number.

## Results

### Loss of CHD-1 partially restores centriole duplication in a *zyg-1* hypomorphic mutant

The existing literature in frogs, flies, mice and humans collectively suggests that CHD-1 plays a role in regulating centriole duplication. Therefore, we first set out to determine if CHD-1 is essential for regulating centriole number by obtaining an existing *chd-1* partial deletion allele from the *Caenorhabditis* Genetics Center (CGC). The *chd-1(ok2798)* allele deletes the C-terminal 197 amino acids encompassing the helicase and DNA-binding domains (**Fig 1A**), and thus

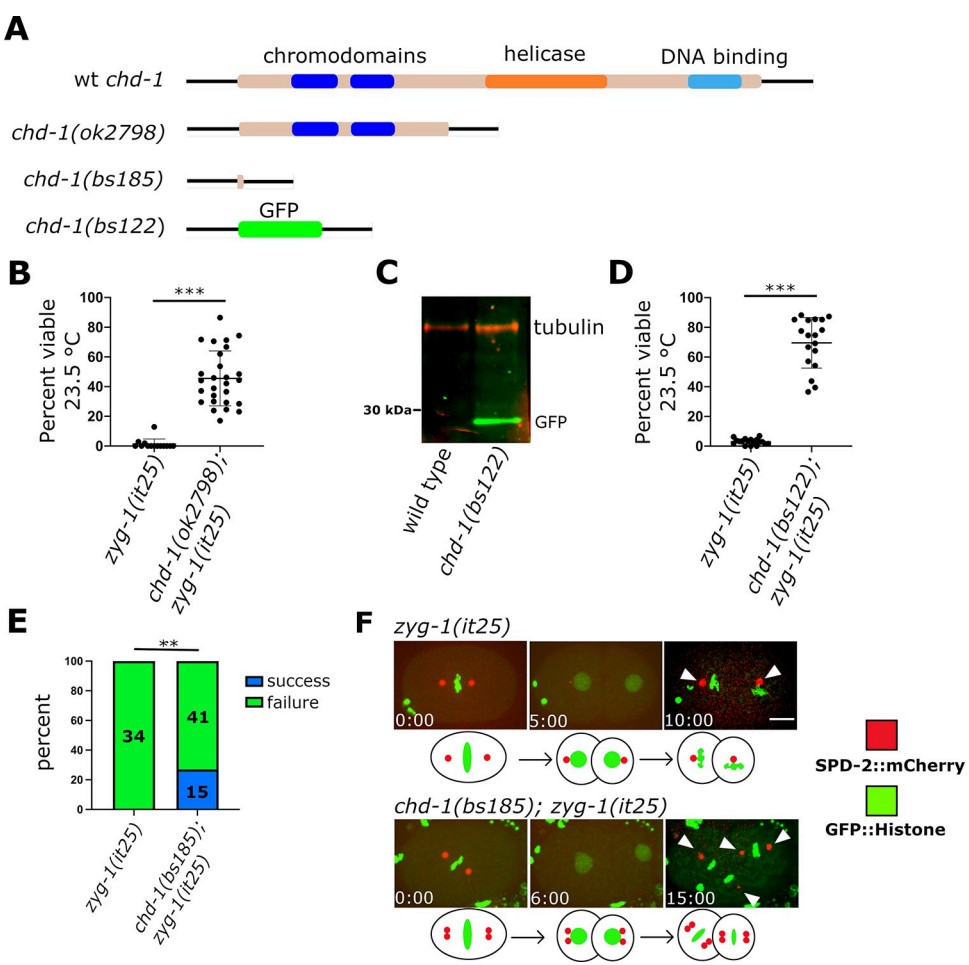

**Fig 1. Deletion of the *chd-1* gene partially rescues the embryonic lethality and centrosome duplication defect of the *zyg-1(it25)* mutant.** (A) Schematic of the wild-type CHD-1 protein and protein products of alleles used in this study. (B) Suppression of embryonic lethality of *zyg-1(it25)* mutants by the original allele *chd-1(ok2798)*. Each data point represents the percentage of viable progeny from a single hermaphrodite. Bars indicate mean and standard deviation. ***p < 0.0001 as calculated by Chi squared analysis (C) An immunoblot showing that the *chd-1(bs122)* allele drives expression of GFP. (D) The *chd-1(bs122)* allele partially suppresses the embryonic lethality of the *zyg-1(it25)* mutant. Each data point represents the percentage of viable progeny from a single hermaphrodite. Bars indicate mean and standard deviation. ***p < 0.0001 as calculated by Chi squared analysis. (E) Loss of CHD-1 partially suppresses the centrosome duplication defect of the *zyg-1(it25)* mutant. The graph depicts the percentage of centrosome duplication failures or successes as determined by live cell imaging of *zyg-1(it25)* and *chd-1(bs185); zyg-1(it25)* embryos expressing GFP::histone and SPD-2::mCherry. Number in parenthesis indicate number of events scored. **p < 0.001 as calculated by Fisher's Exact Test. (F). Frames from select time-lapse imaging data sets of *zyg-1(it25)* and *chd-1(bs185); zyg-1(it25)* embryos expressing GFP::Histone and SPD-2::mCherry. Arrowheads indicate centrosomes. Elapsed time is shown in minutes:seconds. Scale bar = 10 μm. Below each imaging sequence is a graphic depicting the expected centriole behavior in each strain. Red dots represent individual centrioles.

most likely represents a strong loss of function, if not a molecular null. Animals homozygous for the *chd-1(ok2798)* allele are viable, fertile and lack any obvious morphologic or behavioral defects. Further, nearly all progeny produced by homozygous mutant mothers are viable (98.5% viable n = 12 hermaphrodites, 3291 embryos). This is in contrast to all known mutations that block centriole duplication; in such cases, mutants exhibit a maternal-effect embryonic lethal phenotype and/or a sterile uncoordinated phenotype due to cell division failures in the embryonic or postembryonic tissues respectively [55]. The absence of embryonic lethal and sterile phenotypes indicates that CHD-1 is non-essential in worms, and thus, is not required for centriole duplication.

While our results show that CHD-1 is not a key component of the centriole duplication pathway, it still might play an accessory role. To investigate this, we asked if the loss of CHD-1 could modify the phenotype of a *zyg-1(it25)* mutant. The *zyg-1(it25)* allele is a hypomorphic missense mutation that confers a temperature-sensitive block to centriole assembly in early embryos; as a result, at the restrictive temperature of 24°C, 100 percent of the embryos die. To determine if CHD-1 plays a role in centriole duplication, we first monitored the embryonic viability of *zyg-1(it25)* mutants and *chd-1(ok2798); zyg-1(it25)* double mutants grown at the semi-restrictive temperature of 23.5°C. At this temperature, *zyg-1(it25)* mutants exhibited a very low level of embryonic viability (**Fig 1B**). In contrast, about half of the progeny of the *chd-1(ok2798); zyg-1(it25)* double mutants were viable. These data indicate a potential role for CHD-1 in regulating centriole duplication. We then analyzed spindle assembly in the *zyg-1 (it25)* single and *chd-1(ok2798); zyg-1(it25)* double mutants by DIC live imaging (**S1A Fig**). As previously established, in *zyg-1(it25)* mutants centriole duplication invariably fails (n = 12 centrosomes). As a consequence, the two sperm-derived centrioles direct assembly of a bipolar spindle during first division but each is only able to assemble a monopolar spindle at the two-cell stage. In contrast bipolar spindles assembled 64% of the time in two-cell *chd-1(ok2798); zyg-1(it25)* double mutant embryos (n = 14 centrosomes). Thus, loss of CHD-1 activity partially restores bipolar spindle formation in a ZYG-1-deficient strain.

While the *chd-1(ok2798)* deletion is almost certainly a null allele, it still retains the N-terminal chromodomains. To unambiguously establish the null phenotype and to investigate CHD-1 function further, we created two knock out alleles using CRISPR-Cas9 genome editing. We first replaced the *chd-1* open reading frame with superfolder GFP (sfGFP), allowing us to both verify the complete loss-of-function phenotype and to determine the tissue-specific expression pattern of CHD-1. We refer to this allele as *chd-1(bs122)*. Western blotting of *chd-1(bs122)* worm extracts with an anti-GFP antibody detected a single band close to 30 kDa, the approximate molecular weight of sfGFP, indicating that the 5' and 3' flanking regions of *chd-1* are sufficient to drive sfGFP expression (**Fig 1C**). The second allele, *chd-1(bs185)*, deletes nearly the entire *chd-1* ORF, leaving a sequence that encodes just the first nine and last two amino acids; this allele was used in imaging experiments where the GFP produced by the *chd-1(bs122)* allele would have complicated image analysis. Like *chd-1(ok2798)* animals, *chd-1(bs122)* animals are fertile and wild-type in appearance, and they do not exhibit embryonic lethality when grown over a range of temperatures (**S1B Fig**). We also performed time-lapse imaging of wild-type and *chd-1(bs185)* embryos expressing *gfp::histone* and *spd-2::mCherry* transgenes to determine if loss of CHD-1 caused sublethal defects at the cellular level. Embryos lacking CHD-1 appeared normal in all respects including the timing of early cell division events. At 23.5°, the average time for wild-type and *chd-1(bs185)* embryos to progress from first metaphase to metaphase in the AB blastomere was nearly identical (15.3±1.0 minutes, n = 6 and 15.0 ±1.0 min, n = 5, respectively). Likewise, the average time from first metaphase to metaphase in the P1 blastomere was 17.2±0.8 minutes (n = 6) in the wild type versus 17+0.7 minutes (n = 5) in *chd-1(bs185)* mutants. Surprisingly however, we did find that animals lacking CHD-1 exhibit a

cold-sensitive reduction in brood size (**S1C Fig**). At 25˚C, both wild-type and *chd-1(bs122)* animals produce on average approximately 300 offspring. However, at 20˚C where wild-type animals produce an average of 310 offspring, *chd-1(bs122)* animals produce only an average of 255. This difference is exacerbated at 16˚C where wild-type hermaphrodites produce an average of 303 offspring and *chd-1(bs122)* hermaphrodites produce only an average of 195. Thus CHD-1 is required for normal fecundity at low temperatures.

We next tested how a complete loss of *chd-1* affects the embryonic lethality and centrosome duplication defects of a *zyg-1(it25)* mutant. When introduced into a *zyg-1(it25)* strain, the *chd-1 (bs122)* allele suppressed the embryonic lethality observed at semipermissive temperature (**Fig 1D**); while *zyg-1(it25)* single mutants produce on average only three percent viable offspring, an average of 70 percent of the offspring of *chd-1(bs122); zyg-1(it25)* double mutants survive. To monitor centriole duplication, we performed time-lapse spinning disk confocal imaging of embryos produced by *zyg-1(it25)* and *chd-1(bs185); zyg-1(it25)* hermaphrodites expressing sfGFP::histone and SPD-2::mCherry (**Fig 1E and 1F**). While all centrioles (n = 34) failed to duplicate in *zyg-1(it25)* single mutant embryos, 27 percent of the centrioles (n = 55) duplicated in *chd-1 (bs185); zyg-1(it25)* double mutant embryos. Thus, a complete loss of CHD-1 partially suppresses both the embryonic lethality and centriole duplication defects of the *zyg-1(it25)* mutant, indicating that the presence of CHD-1 somehow normally inhibits the ability of centrioles to duplicate.

## CHD-1 is a broadly expressed nuclear protein that is down-regulated in oocytes and embryos

To gain insight into how CHD-1 might regulate centriole duplication, we sought to determine its expression pattern and subcellular distribution. We first analyzed *chd-1(bs122)* worms, where sfGFP expression is entirely controlled by the *chd-1* cis-acting elements (promoter and 3' utr). We observed broad expression throughout worms including in the germ line and somatic tissues (**Fig 2B**, top panel). The broad expression pattern in worms is consistent with that of the human ortholog which was found to be expressed in many different tissues [38]. Interestingly, while sfGFP was evenly diffuse in somatic cells and the distal germ line, it concentrated strongly in the nuclei of oocytes and early embryonic cells. While we do not currently understand why sfGFP concentrates in these specific nuclei, our results clearly show that the *chd-1* gene is broadly expressed in the worm.

We next fused sfGFP to the 3' end of the endogenous *chd-1* orf using CRISPR-Cas9 genome editing (**Fig 2A**). This allele which we named *chd-1(bs125)* expressed a single protein of the expected size on immunoblots, although the band intensity is very dim suggesting that CHD-1 is a low abundance protein (**Fig 2C**). To determine if CHD-1::sfGFP was functional, we crossed this allele into the *zyg-1(it25)* background and monitored embryonic viability. If the sfGFP tag interferes with the function of CHD-1, we would expect that *chd-1(bs125)* would behave as a loss-of-function allele and suppress the embryonic lethality of the *zyg-1(it25)* mutant. However, this was not the case, as *chd-1 (bs125); zyg-1(it25)* animals exhibited low levels of embryonic viability (**Fig 2D**) that were comparable to those of the *zyg-1(it25)* single mutant (**Fig 1B**). We imaged CHD-1::sfGFP worms and found that CHD-1::sfGFP was enriched in nuclei throughout the worm (**Fig 2B**, bottom panel). This included all germ-line nuclei and many somatic cells such as the intestinal cells and cells of the pharynx and somatic gonad. At a subcellular level, the only structure in which we detected CHD-1 enrichment was the nucleus. Importantly, we did not detect localization to centrosomes. While CHD-1::sfGFP expression was detected in oocyte nuclei, the levels were lower than other germ-line nuclei. Interestingly, we could not detect expression in early embryos (**Fig 2B**, bottom panel), suggesting that CHD-1 does not directly regulate centriole duplication in the embryo but instead may function in the maternal germ line to control gene expression.

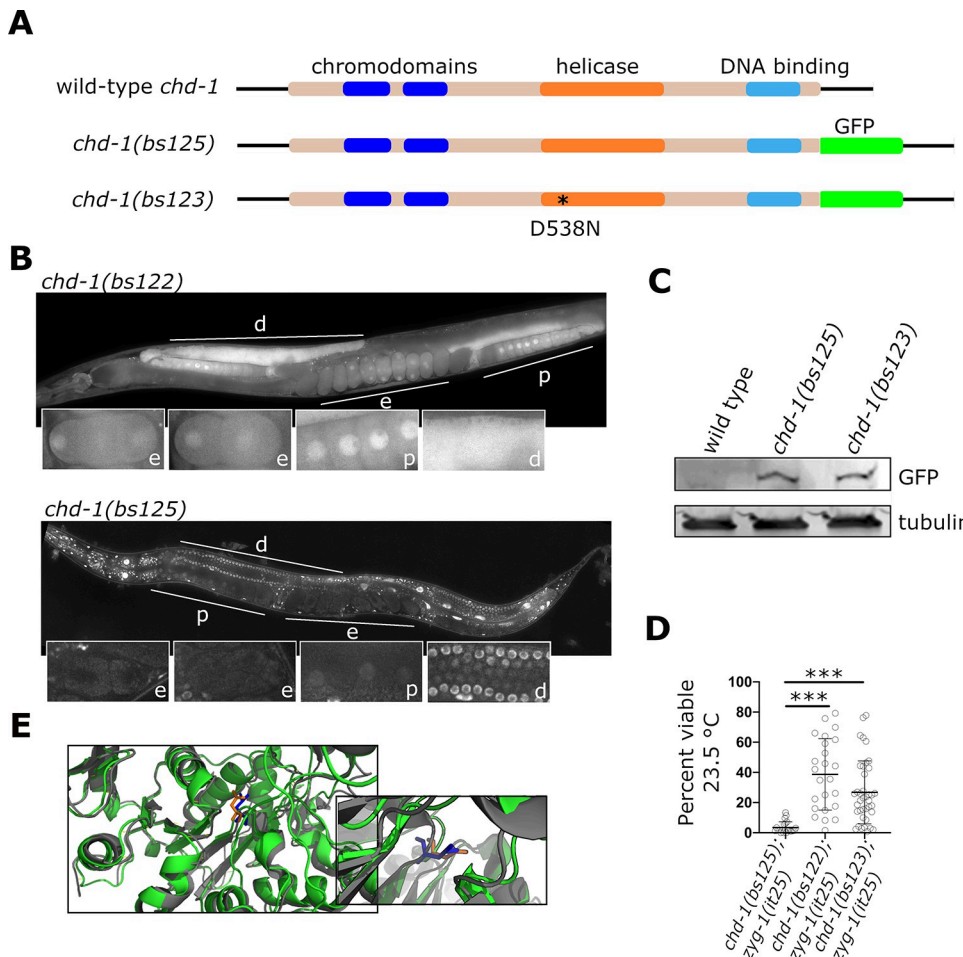

**Fig 2. The *chd-1* gene is broadly transcribed in *C. elegans*, but its protein product is down-regulated in the proximal germ line and early embryo.** (A) Schematic of the protein products of the wild-type *chd-1* gene and two gfp-tagged alleles. The *chd-1(bs125)* allele encodes a wild-type CHD-1 protein C-terminally tagged with GFP. The *chd-1 (bs123)* allele is identical to *chd-1(bs125)* except that it possesses a missense mutation in the helicase domain. (B) Images of hermaphrodites expressing GFP (top) and CHD-1::GFP (bottom) driven by the 5' and 3' *chd-1* flanking regions. Both GFP and CHD-1::GFP are broadly expressed in the soma and germ line but unlike GFP, CHD-1::GFP is concentrated in nuclei throughout all tissues except those of the proximal germ line and early embryo. The distal germ line (d), proximal germ line (p), and early embryos (e) are indicated. Insets show enlarged view of these tissues. (C) Immunoblot probed with an anti-GFP antibody showing that the wild-type and mutant CHD-1::GFP proteins are expressed at about the same level. (D) CHD-1::GFP but not CHD-1(D538N) is functional. The *chd-1(bs125)* and *chd-1 (bs123)* alleles were tested for suppression of the *zyg-1(it25)* embryonic lethal phenotype. Unlike the null *chd-1(bs122)* allele, the *chd-1(bs125)* allele fails to suppress *zyg-1(it25)*, demonstrating that CHD-1::GFP is functional. The *chd-1 (bs123)* allele suppresses *zyg-1(it25)* indicating that helicase activity is essential for regulating centriole duplication. ***p < 0.0001 as calculated by an unpaired t test with Welch's correction. (E) Superimposition of the predicted structure of *C. elegans* CHD-1 protein (green) with that of the *S. cerevisiae* Chd1 protein (gray, accession 3MWY). The inset shows an enlargement focusing on the region proximal to aspartate 513 of yeast Chd1 (blue) aligned with aspartate 538 of worm CHD-1 (orange).

## The helicase activity of CHD-1 is required for regulation of centriole duplication

Next, we sought to determine the molecular mechanism by which CHD-1 regulates centriole duplication. Since *Xenopus* CHD-1 has been reported to physically interact with the frog ZYG-1 homolog [53], we used Immunoprecipitation (IP) (**S2A Fig**) followed by mass spectrometry to

determine if such an interaction exists in worms. Although we identified several proteins that were specifically enriched in the CHD-1 immunoprecipitate (**S2B Fig**), including the putative worm ortholog of CNBP, an RNA Pol II regulator, we were unable to identify an interaction between CHD-1 and any of the known centriole assembly factors including ZYG-1. These results are consistent with CHD-1 regulating centriole duplication through a more indirect mechanism.

The best characterized function of CHD-1 is that of a chromatin remodeling protein. Therefore, we wondered whether CHD-1 could affect centriole duplication by modulating gene expression. As a means to address this question, we sought to engineer an ATPase-inactive version of CHD-1, since the ability of CHD-1 to remodel chromatin is ATP dependent [56]. As the crystal structure of *C. elegans* CHD-1 is not available, we predicted *C. elegans* CHD-1 structure using Swiss Model and aligned it to the available crystal structure of yeast CHD-1 [57]. From this alignment we identified residue D538 of worm CHD-1 as corresponding to aspartate 513 within the Walker B motif of yeast CHD1 (**Fig 2E**). This residue is critical for activation of the helicase ATPase [57]. We then used CRISPR-Cas9 genome editing to create the *chd-1(bs123)* allele, which encodes CHD-1(D538N)::sfGFP (**Fig 2A**). To determine if the ATPase activity of CHD-1 is important for its function in centriole duplication, we created a *chd-1(bs123); zyg-1(it25)* double mutant. Significantly, we found that expression of the ATPase inactive CHD-1 mutant suppressed the embryonic lethal phenotype of *zyg-1(it25)* mutants, indicating that the ATPase activity of CHD-1 is important for its role in regulating centriole duplication (**Fig 2D**). The ability of the *chd-1(bs123)* mutation to suppress *zyg-1(it25)* was not due to altered expression of the ATPase-inactive form of CHD-1::GFP as quantitative western blotting indicated that CHD-1(D538N):: sfGFP and CHD-1::sfGFP are expressed at similar levels (**Fig 2C**). Suppression was also not the result of mis-localization of CHD1(D538N)::sfGFP, as the CHD-1(D538N)::sfGFP protein exhibited a similar localization pattern to that of wild-type CHD-1::sfGFP (**S3 Fig**). Collectively, these data indicate that CHD-1 ATPase activity is specifically required for its role in regulating centriole duplication, suggesting that CHD-1 most likely functions to regulate transcription.

## Loss of CHD-1 results in the upregulation of SAS-6 and SPD-2 protein levels

One mechanism through which loss of CHD-1 activity might suppress the *zyg-1(it25)* centriole duplication defect could involve upregulation of centriole assembly factors, as we previously showed that *zyg-1(it25)* strains expressing elevated levels of ZYG-1 itself, or SAS-6 are able to duplicate their centrioles [27,58]. We therefore sought to determine if the levels of ZYG-1, SPD-2, SAS-5 or SAS-6 were altered in worms carrying the *chd-1(bs122)* null allele. As ZYG-1 is difficult to detect on immunoblots, we used quantitative immunofluorescence to measure ZYG-1 levels at centrioles in wild-type and *chd-1(bs122)* mutants; for this we employed an epitope-tagged version of the endogenous *zyg-1* gene (*spot::zyg-1*), and co-stained with anti-SAS-4 to mark centrioles and anti-SPOT nanobody to measure ZYG-1 levels. As shown in **S4A Fig**, the level of ZYG-1 at the centrosome was slightly but significantly elevated in the *chd-1(bs122)* mutant relative to the wild type. While overexpression of *zyg-1* could certainly contribute to the *chd-1*-mediated suppression of *zyg-1(it25)*, the magnitude of this effect suggested that another mechanism was primarily responsible for suppression. We therefore measured the levels of the other centriole assembly factors by quantitative immunoblotting and surprisingly found that SAS-5 protein levels are reduced in *chd-1(bs122)* mutant relative to the wild type (**S4B Fig**). While unexpected, this result rules out the possibility that the suppression observed upon loss of CHD-1 is due to overexpression of SAS-5. Finally, we analyzed SPD-2 and SAS-6 and found that the levels of both proteins are elevated in *chd-1(bs122)* worms (**Fig 3A** and **3B**). SPD-2 was found to be elevated almost two-fold in the absence of CHD-1, while SAS-6 was

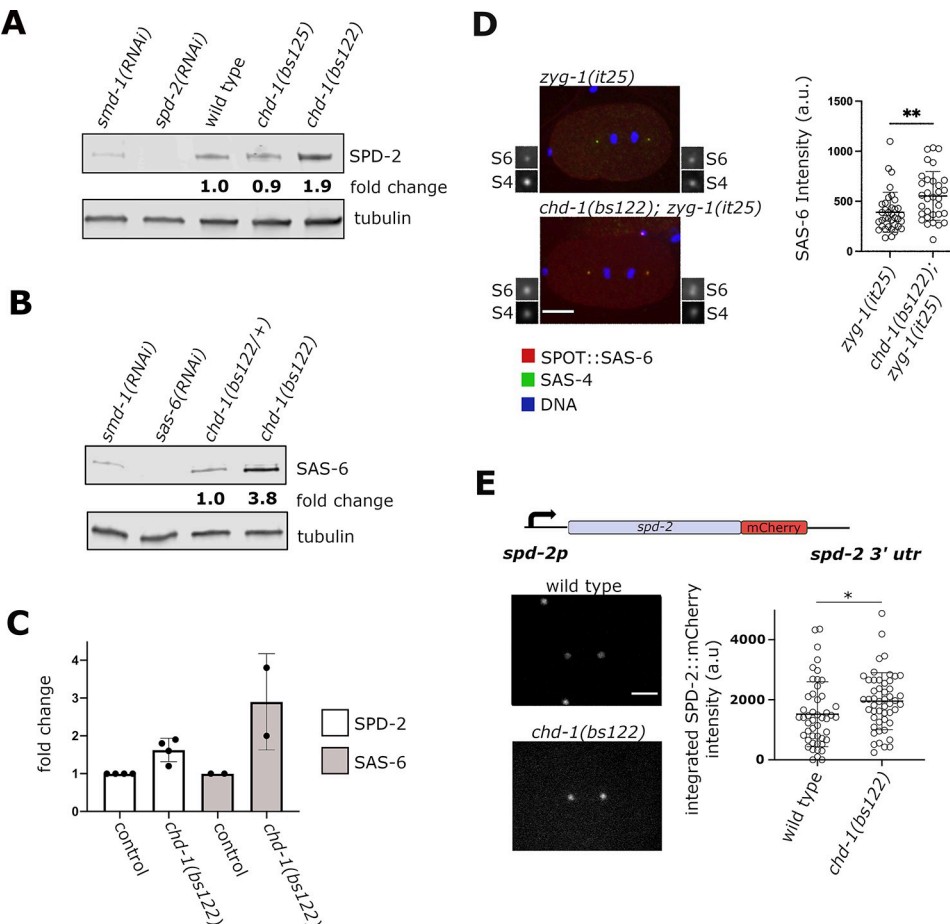

**Fig 3. Loss of CHD-1 results in over-expression of SPD-2 and SAS-6.** (A) Quantitative immunoblot comparing SPD-2 levels in wild-type and *chd-1* mutants. Note the nearly two-fold increase in the *chd-1(bs122)* deletion strains as compared to the wild-type strain. (B) Quantitative immunoblot comparing SAS-6 levels in control (*chd-1(bs122)*/+ heterozygotes) and *chd-1(bs122)* homozygotes. (C) Quantitation of SPD-2 and SAS-6 levels as determined by immunoblotting. (D) *zyg-1(it25); sas-6(bs188(spot::sas-6)* and *chd-1(bs122); zyg-1(it25); sas-6(bs188(spot::sas-6)* embryos immunostained for SPOT::SAS-6 (red), SAS-4 (green) and DNA (blue). Each image is a maximum intensity projection of 24 focal planes (scale bar = 10 μm). Boxes are 3-fold magnified images of centrosomes showing SAS-6 (S6) and SAS-4 (S4) channels. Quantification of SAS-6 staining in shown on right (a.u., arbitrary units). Each data point represents a single centrosome. Bars indicate mean and standard deviation. **p<0.01, unpaired t-test. (E) Quantitative fluorescence microscopy of a *spd-2::mCherry* transgene in heterozygous and homozygous embryos. As shown in the schematic, the transgene is expressed under control of the native *spd-2* promoter and 3' utr, and its expression is positively affected by loss of CHD-1. Representative images (left, scale bar = 10 μm) and quantitation (right) are shown. Each dot represents a single centrosome. Bars indicate mean and standard deviation. *p<0.05, unpaired t-test with Welch's correction.

elevated approximately three-fold (**Fig 3C**). We also found that SPD-2 levels were unaffected in the *chd-1(bs125[chd-1::sfgfp])* strain consistent with our finding that CHD-1::sfGFP is functional (**Fig 3A**). Thus, the two known ZYG-1-binding proteins, SPD-2 and SAS-6, are expressed at higher levels in *chd-1* null mutants.

## The level of SPD-2 at centrosomes is elevated in the absence of CHD-1

To confirm and extend these results, we examined the levels of SAS-6 and SPD-2 at centrosomes in control and *chd-1(bs122)* embryos. We first examined endogenous SAS-6 by immunofluorescence in an isogenic pair of wild-type and *chd-1(bs122)* strains carrying a spot-tagged

version of the *sas-6* gene. Embryos were co-stained with anti-SAS-4 antibody to mark centrioles and anti-SPOT nanobody to measure SAS-6 levels. As shown in **S5A Fig**, we detected a slight increase in the average intensity of SPOT::SAS-6 at centrosomes in the *chd-1(bs122)* mutant relative to controls. However, this increase did not prove to be significant, indicating that the increase in overall levels of SAS-6 in the *chd-1(bs122)* mutant does not necessarily translate into significantly higher levels of SAS-6 at the centrosome. In this experiment we also quantified the centrosome-associated level of SAS-4, a downstream component of the centriole assembly pathway, and found there to be no difference between wild-type and *chd-1(bs122)* mutants (**S5A Fig**).

Our inability to observe an increase in the level of SAS-6 at centrosomes in the *chd-1(bs122)* mutant might be explained by the fact that in the strains analyzed, the centriole assembly pathway is fully functional. In this circumstance, the 3-fold increase in total levels of SAS-6 might not be enough to drive the amount of centrosome-associated SAS-6 much higher. Thus, we chose to re-examine the effect of loss of CHD-1 on centrosome-associated SPOT::SAS-6 in a sensitized background. For this experiment we used strains containing the *zyg-1(it25)* mutation. Remarkably, we found that centrosome-associated SPOT::SAS-6 is increased 1.4-fold in *chd-1 (s122); zyg-1(it25)* embryos as compared to control *zyg-1(it25)* embryos (**Fig 3D**). Thus, in a *zyg-1(it25)* background, the *chd-1(bs122)* mutation leads to an increase in the total SAS-6 level, an increase in the centrosome-associated SAS-6 level, and restoration of centriole assembly.

For measuring SPD-2 levels at the centrosome, we initially employed a *gfp*::*spd-2* transgene driven by the *pie-1* promoter and 3'utr. Contrary to our expectations, we found that GFP::SPD-2 fluorescence intensity at the centrosome was strongly reduced in *chd-1(bs122)* homozygotes compared to their heterozygous siblings (**S5B Fig**). In fact, most (58%, n = 52) of the *chd-1(bs122)* embryos lacked detectable GFP::SPD-2 expression. When these same worms were analyzed by quantitative immunoblotting, we found a reduction in the level of GFP::SPD-2 protein and a corresponding increase in the level of endogenous SPD-2 (**S5C Fig**). We suspect two independent mechanisms that regulate SPD-2 expression account for this result. First, CHD-1 likely inhibits SPD-2 expression directly or indirectly through the 5'- and/or 3'-flanking regions of the *spd-2* gene. Since these elements are absent in the *gfp*::*spd-2* transgene, loss of CHD-1 would not be expected to elevate GFP::SPD-2 expression. Second, as shown by Decker et al., [59], *C. elegans* embryos possess a homeostatic mechanism that helps maintain SPD-2 levels within a specified range. Thus, we envision that loss of CHD-1 relieves an inhibitory mechanism that operates through the *spd-2* flanking regions resulting in over-expression of the endogenous gene. In response to increased SPD-2, the homeostatic control mechanism attempts to decrease overall SPD-2 levels. This set of events would result in a net increase of endogenous SPD-2 and a net decrease of GFP::SPD-2 in the *chd-1(bs122)* strain. To test this idea, we analyzed the levels of a *spd-2*::*mcherry* transgene in control and *chd-1(bs122)* embryos; in contrast to the *gfp*::*spd-2* transgene, expression of the *spd-2*::*mcherry* transgene is driven by the endogenous *spd-2* flanking regions. As shown in **Fig 3E**, the response of this transgene to the absence of CHD-1 was distinctly different from the *pie-1*-driven construct; relative to controls, SPD-2::mCherry fluorescence at the centrosome is elevated in *chd-1(bs122)* embryos. We conclude that CHD-1 inhibits expression of SPD-2 in a manner that is dependent on the *spd-2* promoter and/or 3' utr. Further, our results also demonstrate that homeostatic control of SPD-2 is independent of the 5'- and 3'-flanking regions.

## CHD-1 does not regulate *sas-6* or *spd-2* transcript levels

So far, our work shows that SAS-6 and SPD-2 proteins are significantly upregulated in the absence of CHD-1. To determine if CHD-1 controls the abundance of *sas-6* or *spd-2* transcript

levels, and to determine what other genes might be regulated by CHD-1, we performed RNA-Seq on whole adult worms. Consistent with studies in yeast [60,61], we found that in *C. elegans* only a few genes were strongly affected by loss of CHD-1; specifically, expression of just 11 genes differed by two-fold or more between *chd-1(bs122)* and wild-type worms (**Fig 4A**), but none of the genes in this set are known to function in centriole assembly. We therefore further analyzed this data set by assigning a p value of less than 0.05 as a cut off and found that the transcript levels of approximately 2200 genes were weakly altered in the mutant (**S1** and **S2 Tables**). Surprisingly, none of the genes encoding centriole duplication factors were among this group. To confirm our RNA-Seq results, we used qRT-PCR to compare the levels of *sas-6* and *spd-2* transcripts in wild-type and *chd-1(bs122)* worms. Consistent with our RNA-Seq results, qRT-PCR analysis revealed that loss of CHD-1 did not affect the transcript levels of either gene (**Fig 4B**). Together, our data indicate that CHD-1 does not regulate expression of *spd-2*, *sas-6*, or any other known core centriole assembly factor at the level of transcription or message stability.

## CHD-1 and EFL-1/DPL-1 function independently to control SAS-6 abundance and centriole number

As shown in a prior study, overexpression of SPD-2 is not sufficient to suppress *zyg-1(it25)* embryonic lethality [58]. We thus reasoned that the elevated level of SAS-6 observed in *chd-1* mutants is primarily responsible for rescue of the *zyg-1(it25)* centriole duplication defect. This idea is consistent with our prior work demonstrating that loss of the heterodimeric transcription factor EFL-1/DPL-1 results in the upregulation of SAS-6 protein levels and suppression of *zyg-1(it25)* [27]. Interestingly, our prior work also showed that EFL-1/DPL-1, like CHD-1, does not repress SAS-6 expression by regulating *sas-6* transcript levels. We thus wondered if CHD-1 and EFL-1/DPL-1 function in the same pathway to regulate SAS-6 and centriole duplication. If so, the effect of knocking out both transcription factors should not be greater than the effect of knocking out either factor individually. Therefore, we asked if depleting DPL-1 would enhance suppression of the embryonic lethal phenotype in a *chd-1(bs122); zyg-1(it25)* strain. We performed this assay at 24° where *chd-1(bs122)* suppression was relatively weak, so that any enhancement of suppression could be easily detected. As shown in **Fig 5A**, under our conditions, RNAi of *dpl-1* in either a wild-type or *zyg-1(it25)* mutant background did not alter the level of embryonic viability relative to RNAi of the nonessential gene *smd-1*, which served as a negative control. Since, it has previously been established that loss of DPL-1 significantly decreases embryonic viability in wild-type worms [36,44] and increases embryonic viability in *zyg-1(it25)* worms [29,34], the lack of an effect in either genetic background indicates that RNAi-based depletion of DPL-1, as employed here, is weak. Despite this, when *chd-1(bs122); zyg-1(it25)* worms were treated with *dpl-1(RNAi)*, they exhibited a significant boost in embryonic viability compared to controls (52% vs 6% respectively; **Fig 5A**). Thus, loss of EFL-1/DPL-1 enhances the ability of a *chd-1* null allele to suppress *zyg-1(it25)*, arguing that these two factors likely function independently to control centriole duplication.

Our results show that loss of either CHD-1 or EFL-1/DPL-1 leads to increased levels of SAS-6. Since our genetic analysis suggests that these factors function in different pathways, we hypothesized that loss of both factors would lead to an even higher level of SAS-6 that could result in centriole amplification as seen when SAS-6 is overexpressed in other systems [20,21,62]. To address this possibility, we investigated if *chd-1* and *dpl-1* genetically interact in an otherwise wild-type background. As shown in **Fig 5B**, RNAi-based depletion of DPL-1 in a wild-type background had no effect on embryonic viability (compare *smd-1(RNAi)* vs *dpl-1(RNAi)*). However, in a *chd-1(bs122)* background, *dpl-1(RNAi)* led to a dramatic decrease in

# A

| Genes strongly affected by loss of CHD-1 | | |
|---|---|---|
| Gene | Log₂ (Fold Change) | Expression Ratio (*chd-1*(Δ)/wt) |
| *clec-47* | 1.427 | 2.7 |
| *pgp-7* | 1.356 | 2.6 |
| *pgp-6* | 1.223 | 2.3 |
| *sodh-1* | 1.103 | 2.1 |
| *gst-4* | 1.084 | 2.1 |
| C26H9A.2 | 1.038 | 2.1 |
| *gtl-1* | 1.038 | 2.1 |
| K10C2.3 | 1.003 | 2.0 |
| *plk-3* | -2.155 | 0.25 |
| *fbxa-192* | -1.933 | 0.26 |
| *sax-2* | -1.362 | 0.38 |

# B

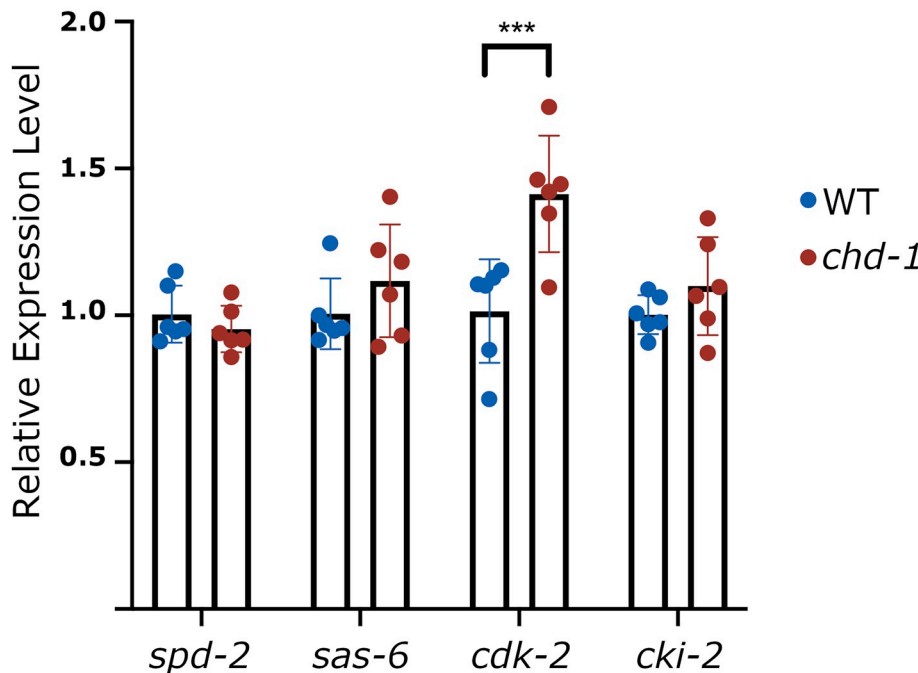

**Fig 4. *cdk-2* transcript levels are elevated in *chd-1(bs122)* mutants.** (A) List of genes whose mRNA levels are affected by two-fold or greater by loss of CHD-1 as determined by RNA-Seq. (B) Results of qRT-PCR of adult wild-type and *chd-1(bs122)* worms showing relative transcript levels. ***p<0.0001, t-test with a Holm-Sidak correction.

embryonic viability with about half of the embryos failing to hatch (compare *chd-1; smd-1 (RNAi)* to *chd-1; dpl-1(RNAi)*). Thus, *chd-1* and *dpl-1* genetically interact.

As mentioned above, *dpl-1(RNAi)* is inefficient. Thus, we wondered if a stronger genetic interaction might be identified using a loss-of-function allele of *dpl-1*. The *dpl-1(bs21)* allele

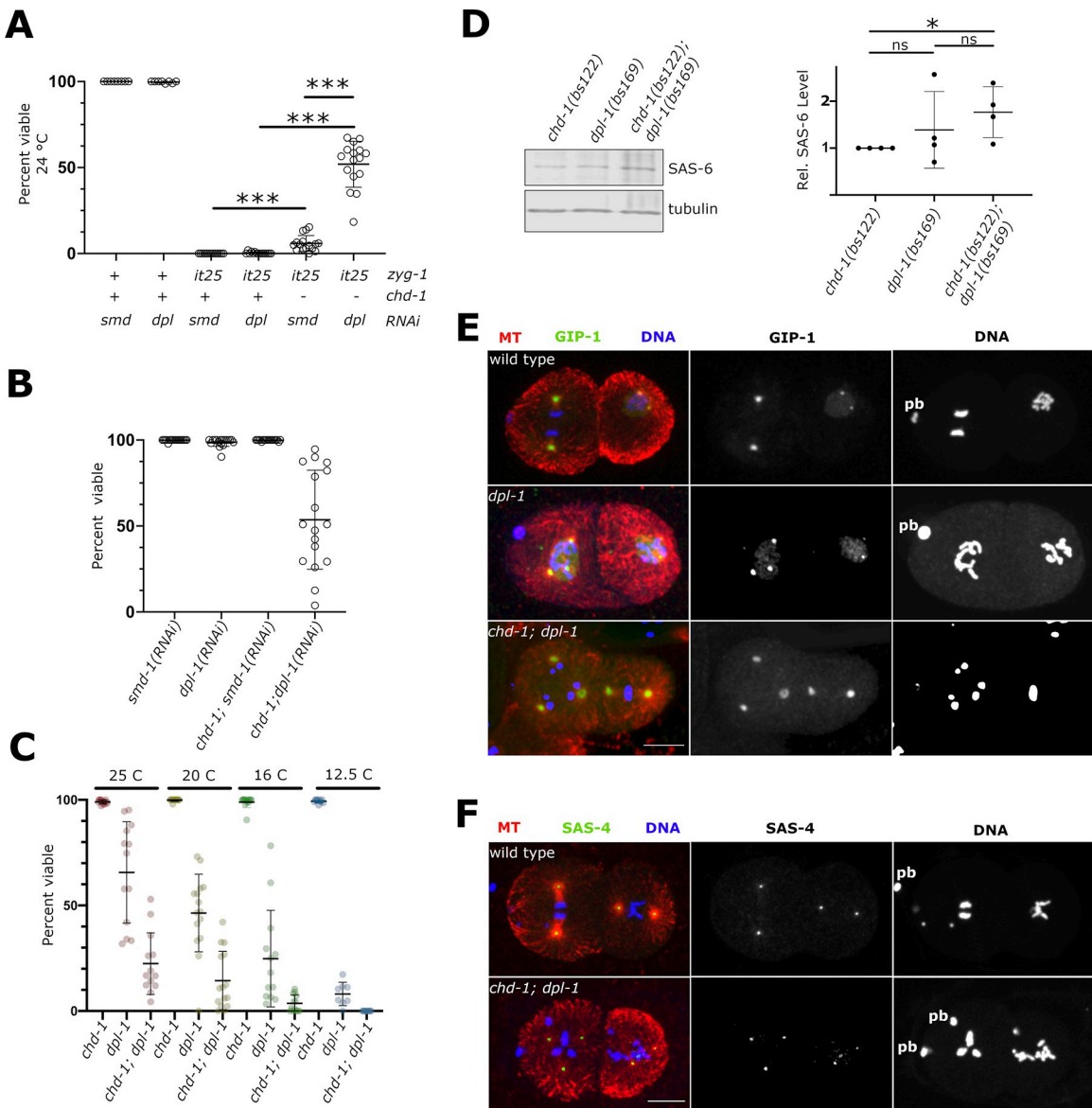

**Fig 5. CHD-1 and DPL-1 cooperate to control SAS-6 levels and proper centriole number.** (A) Suppression of the embryonic lethal phenotype of *zyg-1(it25)* at 24° in the presence of the *chd-1(bs122)* allele and/or *dpl-1(RNAi)*. Wild-type alleles are indicated by a plus sign while mutant alleles are indicated by either *it25* or a minus sign. Each data point represents the percentage of viable progeny from a single hermaphrodite. Bars indicate mean and standard deviation. Note that the level of suppression observed in the presence of both *chd-1(bs122)* and *dpl-1(RNAi)* is greater than the expected level of suppression if the effects of *chd-1(bs122)* and *dpl-1(RNAi)* were additive. ***p<0.0001, Welch's t test. (B) The simultaneous depletion of CHD-1 and DPL-1 results in embryonic lethality. *dpl-1(RNAi)* or control RNAi (*smd-1*) were performed in a wild-type or *chd-1(bs122)* background. Specifically, the combination of *chd-1(bs122)* and *dpl-1(RNAi)* led to embryonic lethality. Bars indicate mean and standard deviation. (C). The *chd-1(bs122)* deletion allele enhances the embryonic lethality of *dpl-1(bs169)*. Embryonic viability was measured at the indicated temperatures in *chd-1(bs122)*, *dpl-1(bs169)*, and *chd-1(bs122)*; *dpl-1(bs169)* strains. While the *chd-1(bs122)* strain was fully viable at all temperatures, the *dpl-1(bs169)* and *chd-1(bs122)*; *dpl-1(bs169)* strains displayed embryonic lethality that became progressively more severe as the temperature was reduced. At all temperatures tested, the *chd-1(bs122)*; *dpl-1(bs169)* double mutant exhibited a more severe phenotype than the *dpl-1(bs169)* single mutant. Bars indicate mean and standard deviation. (D) Quantitative immunoblot of SAS-6 from whole lysates of *chd-1(bs122)*, *dpl-1 (bs169)*, and *chd-1(bs122)*; *dpl-1(bs169)* worms grown at 16˚C. Quantitation of relative SAS-6 levels is shown on the right. Bars indicate mean and standard deviation. n.s. not significant, *p< 0.05, t-test. (E) Wild-type, *dpl-1(bs169)*, and *chd-1(bs122)*; *dpl-1(bs169)* embryos grown at 12.5˚ were immunostained for alpha-tubulin (red), the PCM protein GIP-1 (green) and DNA (blue). Images show maximum intensity projections of two-cell stage embryos. Polar body DNA (pb) is indicated. Scale bar = 10 μm. (F) Wild-type and *chd-1(bs122)*; *dpl-1(bs169)* embryos grown at 12.5˚ were immunostained for alpha-tubulin (red), the centriole protein SAS-4 (green) and DNA (blue). Polar body DNA (pb) is indicated. Images show maximum intensity projections of two-cell stage embryos. Scale bar = 10 μm.

identified in our genetic screen is a nonsense mutation (Q521X) that results in truncation of the last 196 amino acid residues while leaving the DNA-binding and heterodimerization domains intact. The *dpl-1(bs21)* allele behaves as a hypomorph and confers a partial embryonic lethal phenotype [27,63]. Using CRISPR-Cas9 genome editing we recreated this allele which we named *dpl-1(bs169)*. We then quantified the effect of this allele on embryonic viability either alone or in the presence of the *chd-1(bs122)* allele over a range of temperatures (**Fig 5C**). Surprisingly we found that *dpl-1(bs169)* confers a cold-sensitive embryonic lethal phenotype. Hermaphrodites, carrying the *dpl-1(bs169)* mutation exhibit relatively high embryonic viability (66%) at 25˚ but progressively lower embryonic viability as the temperature is decreased (46% at 20˚, 25% at 16˚, and 8% at 12.5˚). Strikingly, hermaphrodites carrying both the *chd-1 (bs122)* and *dpl-1(bs169)* alleles consistently display lower embryonic viability than the *dpl-1 (bs169)* single mutants at all temperatures tested (22% at 25˚, 14% at 20˚, 3% at 16˚, and 0% at 12.5˚). The increased embryonic lethality observed in the double mutant is not due to an additive effect, as the *chd-1(bs122)* allele does not produce an embryonic lethal phenotype on its own at any of the temperatures tested. Thus, the loss of CHD-1 enhances the embryonic lethal phenotype of a *dpl-1* loss-of-function allele.

Our data so far are consistent with the possibility that SAS-6 expression is elevated in the double mutant relative to the single mutants and that this elevated level of SAS-6 is toxic. To directly address this, we performed quantitative immunoblots on lysates prepared from *chd-1 (bs122)*, *dpl-1(bs169)*, and *chd-1(bs122); dpl-1(bs169)* mutant worms grown at 16˚C. As shown in **Fig 5D**, we found that SAS-6 was elevated an average of 1.8-fold in the double mutant, relative to the *chd-1(bs122)* single mutant. We also found a smaller 1.3-fold increase in the double mutant relative to the *dpl-1(bs169)* single mutant, although this difference did not achieve statistical significance. We conclude that SAS-6 is increased in the double mutant relative to the *chd-1* single mutant and possibly relative to the *dpl-1* mutant as well.

Previously, we have shown that a small percentage of embryos produced from *dpl-1(bs21)* hermaphrodites possess extra MTOCs [27]. This analysis however was performed at elevated temperature where the *dpl-1(bs21)* allele is still relatively active. Thus, we sought to determine how loss of either EFL-1/DPL-1 alone or the combined loss of both CHD-1 and EFL-1/DPL-1 affects centriole duplication under the most restrictive of conditions. Single and double mutants were grown at 12.5˚ and embryos immunostained with antibodies to alpha-tubulin and the centrosome marker GIP-1, an otholog of the γ-TuRC component GCP3 [64]. As shown in **Fig 5E**, while wild-type embryos invariably had no more than two GIP-1-positive MTOCs per cell, both *dpl-1(bs169)* single mutant embryos and *chd-1(bs122); dpl-1(bs169)* double mutant embryos contained cells with supernumerary GIP-1-positive MTOCs. Although cytokinesis defects are a known source of extra centrosomes, such a defect does not appear to account for the emergence of extra MTOCs in these mutants. Cytokinesis failure invariably gives rise to an even number of MTOCs whereas we observed numerous cases of cells with an odd number of MTOCs (**Fig 5E**). Furthermore, we did not observe other hallmarks of cytokinesis failure such as binucleate cells (**Table 1**).

Next, we set out to determine if these extra MTOCs represent bona fide centrosomes. We therefore immunostained embryos for the centriole marker SAS-4. As shown in **Fig 5F**, the extra MTOCs in *chd-1(bs122); dpl-1(bs169)* double mutant embryos did indeed contain SAS-4. Furthermore, the relative intensity of SAS-4 staining among the poles of each multipolar spindle was similar to the relative intensity observed between the poles of bipolar spindles from control embryos. That is, within a multipolar spindle, the dimmer poles exhibited on average 81% of the intensity exhibited by the brightest SAS-4-stained pole (n = 4 spindles). This is comparable to the situation in control bipolar spindles where on average the dimmer pole was 67% as intense as the brighter pole (n = 4 spindles). This result suggests that the extra

**Table 1. Loss of *chd-1* enhances the multipolar spindle defect of the *dpl-1(bs169)* mutation.**

| | Genotype | |
|---|---|---|
| | *dpl-1(bs169)* | *chd-1(bs122); dpl-1(bs169)* |
| Total embryos scored | 64 | 70 |
| Total one-cell embryos scored | 18 | 22 |
| Multipolar one-cell embryos | 0 (0%) | 0 (0%) |
| Total 2–6 cell stage embryos scored | 46 | 48 |
| Multipolar 2-6-cell stage embryos | 2 (4.2%) | 18 (27.3%) |
| 2-6-cell stage embryos with binucleate cells | 0 (0%) | 0 (0%) |
| Total cells of 2–6 cell stage embryos | 120 | 125 |
| Multipolar cells of 2–6 cell stage embryos* | 2 (1.6%) | 22 (15%) |

* Chi-squared analysis, p = 0.000027

poles do not arise through fragmentation of centrioles, which would likely result in a significantly greater asymmetry in SAS-4 staining. Thus, the extra centrosomes most likely arise through an over-production mechanism (e.g., overduplication or *de novo* assembly of centrioles).

Analysis of the cellular distribution and frequency of extra centrosomes among the single and double mutant embryos revealed additional details of this defect (**Table 1**). First, no extra centrosomes were observed at the one-cell stage in either single or double mutants. As the zygote's centrioles are of paternal origin, such a defect at this stage of development would indicate a defect in the male germ line. Thus, the absence of such defects suggests that loss of EFL-1/DPL-1 alone or both CHD-1 and EFL-1/DPL-1 selectively affects maternal control of centriole assembly. Second, the *chd-1(bs122)* null allele strongly enhances the extra-centrosome defect of the *dpl-1(bs169)* mutant; only two out of 120 cells from young (two-six-cell stage) *dpl-1(bs169)* embryos possessed one or more extra centrosomes while 22 out of 125 cells from similarly staged *chd-1(bs122); dpl-1(bs169)* embryos exhibited this defect. The difference is highly significant (p<0.0001) indicating that loss of *chd-1* strongly enhances the centrosome amplification defect of the *dpl-1* mutant.

To gain additional insight into the genesis and behavior of these extra centrioles, we performed live imaging of *chd-1(bs122); dpl-1(bs169)* double mutant embryos carrying a *spd-2*::*mCherry* transgene. Double mutant hermaphrodites were shifted to 12.5 C overnight and their offspring recorded at the same temperature the next day. Under these conditions we were able to observe that the extra centrosomes form in the vicinity of other centrioles and only become visible around the time that mother daughter centriole pairs normally separate (**S6 Fig**). These observations are consistent with an overduplication defect where mother centrioles produce more than one daughter centriole. We also observed that the extra centrosomes are capable of duplicating (**S6 Fig**). Thus, in addition to possessing both centriole and PCM components, the extra centrosomes exhibit the behavior of a canonical centrosome.

## CHD-1 and EFL-1/DPL-1 independently down-regulate CDK-2 activity to control SAS-6 abundance

Recent work has shown that EFL-1/DPL-1 down regulates transcription of the *cdk-2* gene in the *C. elegans* germ line [65]. CDK-2 is required for centriole duplication in vertebrates [26,66–68] and has recently been shown to protect the SAS-6 binding partner STIL/SAS-5 from degradation mediated by the $SCF^{\beta TrCP}$ E3 ubiquitin ligase [69]. Therefore, we wondered if CDK-2 might be the relevant target of EFL-1/DPL-1 and possibly CHD-1 as well. Consistent

with such a mechanism, we found that *cdk-2* transcripts are upregulated in the *chd-1(bs122)* mutant relative to the control strain (**Fig 4B**). Thus, both CHD-1 and EFL-1/DPL-1 independently down regulate *cdk-2* expression. We also noticed that *cki-2*, a CDK-2 inhibitor whose down regulation has been reported to result in the presence of supernumerary centrosomes [70], was among the set of genes weakly down regulated in the *chd-1* deletion strain (**S1 Table**). However, qRT-PCR analysis revealed that the level of *cki-2* transcripts is unaffected by the *chd-1(bs122)* mutation (**Fig 4B**). In summary, our data suggest a model whereby the loss of either transcriptional regulator results in elevated levels of CDK-2 activity, which in turn, acts post-transcriptionally to promote SAS-6 expression.

We set out to test our hypothesis by determining if CDK-2 was required for suppression of the embryonic lethality of a *zyg-1(it25)* mutant by loss of either EFL-1/DPL-1 or CHD-1. As demonstrated above, *zyg-1(it25)* animals grown at the nonpermissive temperature of 24˚C do not produce any viable progeny (**Fig 5A**, *zyg-1(it25); smd-1(RNAi)*). In contrast, *zyg-1(it25) dpl-1(bs169)* double mutants grown under identical conditions produce approximately 80% viable offspring (**Fig 6A**), demonstrating the ability of the *dpl-1(bs169)* mutation to robustly suppress the lethality of the *zyg-1(it25)* allele. Strikingly, when the double mutant was treated with *cdk-2(RNAi)* there was almost a complete loss of viability among the offspring of the *zyg-1(it25) dpl-1(bs169)* double mutants. Importantly, depletion of *cdk-2* only mildly affected embryonic viability in wild-type animals, indicating that the nearly complete loss of viability observed in *zyg-1(it25) dpl-1(bs169)* animals exposed to *cdk-2(RNAi)* was due to a loss of suppression rather than disruption of an essential CDK-2-dependent function. We performed the same experiment with the *chd-1(bs122); zyg-1(it25)* double mutant and found that at the semi-permissive temperature of 23.5˚C, *chd-1(bs122)*-mediated suppression of *zyg-1(it25)* was also reduced upon *cdk-2(RNAi)* (**Fig 6B**). However, CDK-2 depletion only partially eliminated suppression by the *chd-1(bs122)* allele, suggesting that CHD-1 might regulate centriole duplication through additional mechanisms that do not involve CDK-2.

We next used quantitative western blotting to determine if CDK-2 controls the level of SAS-6. For these experiments, we employed the auxin inducible degradation system [71] to robustly deplete CDK-2. We therefore constructed a strain harboring an endogenous *cdk-2* gene tagged with the auxin-inducible degron (AID) and carrying a transgene that expresses the substrate recognition component TIR1 in the germ line. This strain was grown in the absence and presence of 1 mM auxin. As shown in **Fig 6C**, upon treatment with auxin SAS-6 dropped to an undetectable level, consistent with a role for CDK-2 in promoting SAS-6 protein levels. Depletion of CDK-2 however does not appear to have a general effect on expression of centriole assembly factors, as the level of SAS-5 is unaffected by auxin-induced degradation of CDK-2 (**Fig 6D**).

In summary, our work strongly supports a model (**Fig 6E**) whereby CHD-1 and EFL-1/DPL-1 function in the maternal *C. elegans* germ line to negatively regulate expression of CDK-2 at the mRNA level. In turn, CDK-2 promotes expression of SAS-6, and when overexpressed can drive the formation of extra centrioles and multipolar spindles.

## Discussion

The cyclin-dependent kinase CDK-2 and members of the E2F family of transcriptional activators have long been known to regulate centriole number in dividing vertebrate cells. In both *Xenopus* eggs and mammalian somatic cells, inhibition of CDK-2 complexed with either cyclin A or cyclin E blocks centriole assembly [26,66–68]. In contrast, E2F members can play either positive or negative roles in centriole biogenesis. DP1, E2F2 and E2F3 have been shown to promote centriole duplication in Chinese hamster ovary cells [26], while loss of E2F3 results in

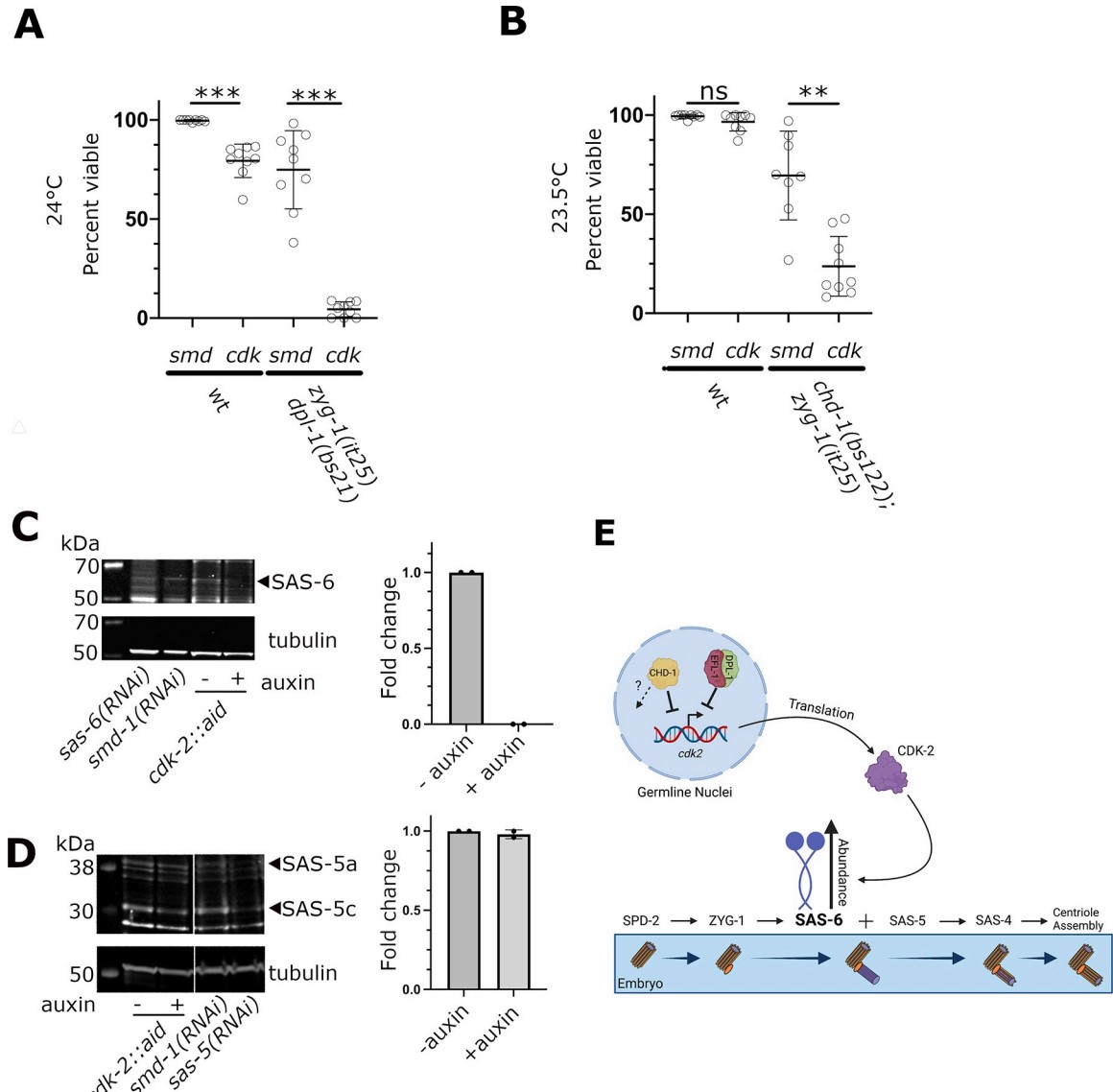

**Fig 6. CDK-2 is required for *dpl-1* and *chd-1*-mediated suppression of *zyg-1*.** (A) Percent embryonic lethality among the progeny of wild-type and *zyg-1(it25) dpl-1(bs21)* strains following *smd-1* (control) and *cdk-2* RNAi. The progeny of wild-type animals subjected to *cdk-2(RNAi)* exhibit a small reduction in viability. In contrast, viability among the progeny of *zyg-1(it25) dpl-1(bs21)* animals subjected to *cdk-2(RNAi)* is almost completely lost, indicating that CDK-2 plays an important role in *dpl-1*-mediated suppression. *** p<0.0001, two-tailed t test with Welch's correction. (B) Percent embryonic lethality in wild-type and *chd-1(bs122); zyg-1(it25)* strains following *smd-1* (control) and *cdk-2* RNAi. The progeny of wild-type animals subjected to *cdk-2(RNAi)* exhibit a small but insignificant reduction in viability. In contrast, viability among the progeny of *zyg-1(it25) dpl-1(bs21)* animals subjected to *cdk-2(RNAi)* is significantly reduced, indicating that CDK-2 plays an important role in *chd-1*-mediated suppression. ** p<0.001, ns = not significant, two-tailed t test with Welch's correction. (C) Left. Immunoblot of whole *cdk-2::aid; TIR-1::mRuby* worms probed for SAS-6 and α-tubulin. The SAS-6 band is identified by its reduction in intensity upon *sas-6(RNAi)*. *smd-1(RNAi)* serves as a negative control. The presence or absence of auxin in the growth media is indicated. Right. Quantitation of SAS-6 levels. (D) Left. Immunoblot of whole *cdk-2::aid; TIR-1::mRuby* worms probed for SAS-5 and α-tubulin. The SAS-5a and SAS-5c bands are identified based on their reduction in intensity upon *sas-5(RNAi)*. *smd-1(RNAi)* serves as a negative control. The presence or absence of auxin in the growth media is indicated. Right. Quantitation of SAS-5a levels. (E) Model of regulatory scheme (created with BioRender.com). In germ line nuclei, EFL-1/DPL-1 and CHD-1 independently down regulate expression of the *cdk-2* gene. CDK-2 protein in turn promotes the expression of SAS-6. Under normal conditions, this pathway ensures that embryos ultimately inherit the proper amount of SAS-6 to support centriole duplication during the embryonic cell divisions. The dashed arrow indicates that CHD-1 also functions in a CDK-2-independent capacity to regulate centriole assembly.

centriole amplification in mouse embryonic fibroblasts [29]. While these pioneering studies hinted that E2F proteins might exert control over centriole assembly by regulating CDK-2 activity, they did not provide definitive proof for such a model. Further, beyond establishing an essential role for CDK-2 in centriole assembly, this early work did not define its mechanism of action.

Our work builds upon these earlier studies by identifying two parallel transcriptional control pathways—one E2F-dependent and one E2F-independent—that converge upon CDK-2 to limit expression of SAS-6 and ensure a precise doubling of centrioles in the *C. elegans* embryo. As shown by Furuta et al., EFL-1/DPL-1 directly down regulates transcription of CDK-2 [65]. Likewise, we find that CHD-1 represses expression of CDK-2, although it is not yet clear if it does so through a direct or indirect mechanism. Nevertheless, our work shows that the two transcriptional regulators fine tune the level of CDK-2 which in turn regulates the abundance of the centriole scaffold protein SAS-6 to ensure centrosome number is properly maintained.

While EFL-1/DPL-1 and CHD-1 both regulate SAS-6 levels, the two pathways differ in two respects. First, the EFL-1/DPL-1-dependent pathway clearly plays the predominant role in regulating centriole assembly. As our data shows, a hypomorphic *dpl-1* allele more strongly suppresses the centriole duplication defect of *zyg-1(it25)* mutants than a complete deletion of *chd-1*. Further, *dpl-1* mutant embryos possess extra centrosomes whereas *chd-1* null embryos do not. A second distinction between the two pathways is the degree to which they depend on CDK-2 for regulating centriole duplication. Upon *cdk-2(RNAi)*, *zyg-1(it25); dpl-1(bs169)* hermaphrodites produce very few viable offspring, indicating that regulation of centriole duplication by EFL-1/DPL-1 is mostly if not entirely dependent on CDK-2 activity. In contrast, a substantial fraction of the offspring of *chd-1(bs122); zyg-1(it25)* hermaphrodites subjected to *cdk-2(RNAi)* remain viable, suggesting that CHD-1 also regulates centriole duplication in a CDK-2-independent manner.

One exciting possibility is that CHD-1 might also function to regulate centriole assembly outside of its role as a chromatin regulator. Hatch et al., [53] reported a physical interaction between Chd1 and PLK4 in Xenopus and we found that ZYG-1 levels are elevated at the centriole in a *chd-1(bs122)* mutant (**S4A Fig**). While we were unable to establish such in interaction in *C. elegans*, we could have missed this if it were a transient interaction or if it only occurred in a specific subset of cells. Intriguingly, there is a precedent for a transcriptional regulator playing a direct role in centriole assembly. In multiciliated cells, E2F4 the ortholog of EFL-1 promotes expression of centriole assembly genes before being translocated to the cytoplasm where it forms apical aggregates containing SAS-6 and deup1, a component of ring-shaped protein complexes called deuterosomes that serve as sites of centriole assembly [28]. Remarkably, mutations that block nuclear export of E2F4 allow transcriptional activation of centriole genes but block deuterosome assembly and centriole amplification [28].

One unexpected twist of this study is the finding that CHD-1 also seems to function as a positive regulator of centriole assembly. Specifically, we found that CHD-1 promotes expression of SAS-5, whose level is significantly diminished in the *chd-1(bs122)* mutant. So how do we reconcile this positive role of CHD-1 with the ability of the *chd-1(bs122)* mutation to restore centriole assembly to a *zyg-1(it25)* mutant or to amplify the multipolar spindle defect of a *dpl-1(bs169)* mutant? SAS-5 and SAS-6 physically interact and are mutually required to localize to the assembling centriole [20]. One possibility is that in *zyg-1(it25)* embryos where centriole assembly is compromised, the level of SAS-5 is not limiting. By increasing the total amount of SAS-6, as well as the centrosome-associated level of ZYG-1, the *chd-1(bs122)* mutation is able to increase recruitment of SAS-6 (**Fig 3D**) and presumably SAS-5, to allow assembly to proceed. In contrast, in a wild-type *zyg-1* background, where the centriole assembly pathway is fully functional, it might be more difficult to drive an increase in SAS-6

recruitment. In this circumstance, the effects of the *chd-1(bs122)* mutation on SAS-6 and ZYG-1 may not be strong enough to compensate for the reduction in SAS-5 level. This would explain why we do not see enhanced SAS-6 recruitment (**S5A Fig**) and multipolar spindles in the *chd-1(bs122)* mutant, despite the elevated level of SAS-6. Finally, in the *chd-1(bs122); dpl-1 (bs169)* double mutant, it is possible that the additional increase in SAS-6 abundance crosses yet another threshold that results in centriole amplification.

Our finding that disruption of both EFL-1/DPL-1 and CHD-1 function leads to elevated SAS-6 levels and the frequent appearance of supernumerary centrosomes, once again highlights how numerical control depends upon the abundance of core centriole assembly factors. Prior work has shown that regulated proteolysis plays a critical role by controlling the levels of ZYG-1/Plk;4, SAS-5/STIL and SAS-6 [7,17,62, 69,72–78]. The basic theme that arises from these studies is that degradation is mediated by an E3 ubiquitin ligase (either SCF or the APC/C), and that temporal control is achieved by regulating substrate recognition through cell-cycle-stage-dependent phosphorylation of either the target protein itself or of the E3 ligase. It therefore seems likely that CDK-2 functions in a similar capacity to control the levels of STIL in vertebrates and SAS-6 in worms. However, in both cases the substrate phosphorylated by CDK-2 remains to be identified.

The question of CDK-2 action notwithstanding, our work, together with that of Arquint et al. 2018 [69], shows that in both worms and humans, CDK-2 regulates centriole assembly by controlling the abundance of centriole scaffold components. In human cells, CDK-2 stabilizes STIL/SAS-5, a binding partner of SAS-6, by blocking degradation via the E3 ubiquitin ligase SCF$^{\beta TrCP}$. In worms, the target of CDK-2 is SAS-6, suggesting an evolutionary divergence of the underlying mechanism. However, the extent to which the mechanisms operating in humans and worms differ is not yet clear. For instance, in humans, SAS-6 levels are also controlled by SCF$^{\beta TrCP}$, but whether CDK-2 also regulates SAS-6 abundance has not been investigated. Conversely in worms, SCF$^{\beta TrCP}$ has been shown to regulate centriole duplication by controlling ZYG-1 levels at the centriole [79], but whether SCF$^{\beta TrCP}$ also regulates SAS-6 is an open question. Thus, it remains possible that the CDK-2-dependent pathways operating in vertebrates and worms are not all that different.

Over the past two decades research has uncovered a strong link between centriole amplification and human diseases including cancer and primary microcephaly. Interestingly E2F proteins have been implicated in both diseases [80,81], although in neither disease has a link between loss of E2F activity and centriole amplification been established. Our work showing that a loss of EFL-1/DPL-1, or the combined loss of EFL-1/DPL-1 and CHD-1 leads to the presence of extra centrosomes in the *C. elegans* embryo is thus relevant to understanding how these diseases might arise in response to a loss of numerical control.

## Materials and methods

### Worm strains and maintenance

Worms were maintained on MYOB agar plates seeded with *E. coli* OP50 at 20˚C according to standard protocols [82]. Strains used in this study are described in **S3 Table**.

### CRISPR-Cas9 Genome editing

CRISPR-Cas9 genome editing was performed by microinjection of *in-vitro* assembled ribonucleoprotein complexes essentially as previously described [83,84]. Screening utilized the co-CRISPR strategy [85]. The crRNAs were purchased from Dharmacon, Inc. (Lafayette, CO) and primers and oligonucleotide repair templates were purchased from Integrated DNA technologies (Coralville, IA). Cas9 protein was purified according to the protocol by Paix et al,

2015 [84]. The crRNA and repair template sequences for all the strains generated by CRISPR-Cas9 editing in this study are provided in **S4 Table**.

### Embryonic viability assays and brood counts

For most measurements of embryonic viability, including all embryonic lethality suppression assays, L3-L4 larvae were picked individually to 35 mm MYOB agar plates at the indicated temperature and allowed to lay eggs for 1 day (20˚C, 25˚C), 2 days (16˚C) or 3 days (12.5˚C). The adult worms were removed, and the plates were incubated at the same temperature for an additional period of time equal to the egg-laying period. The number of dead and live progeny were then manually counted. For brood size measurements, L4 larvae were picked individually to 35 mm MYOB agar plates. These plates were incubated at the indicated temperature and the adult transferred to a fresh plate every 24 hr until egg laying ceased. Live and dead progeny were counted on each plate 1 day (20˚C, 25˚C) or 2 days (16˚C) after removing the adult.

### RNAi

RNAi was administered by feeding as described previously [86]. In brief, animals at the L1 or L4 stage were transferred to a lawn of dsRNA-expressing bacteria that had been grown on MYOB plates supplemented with 50–100 μg/ml carbenicillin, with or without 25 μg/ml tetracycline and 1–2 mM IPTG (Isopropyl β- d-1-thiogalactopyranoside). Plates were then incubated at the specified temperature. RNAi against the nonessential *smd-1* gene served as a negative control.

### Fixed and live imaging

Immunofluorescence microscopy was essentially performed as previously described [87]. The mouse monoclonal anti-alpha-tubulin antibody DM1A (Sigma-Aldrich, St. Louis, MO) and the rabbit anti-GIP-1 antibody [64] were used at a dilution of 1:1000. Alexa 568 anti-mouse and Alexa 488 anti-rabbit secondary antibodies (Thermo Fisher Scientific, Waltham, MA) were used at a 1:1000 dilution. For detection of the SPOT tag, we used the SPOT-Label Alexa-Fluor 568 nanobody (Chromotek, Planegg-Martinsried, Germany) at a 1:1000 dilution. For time-lapse imaging of embryos, worms were grown at the indicated temperature and embryos dissected and mounted as described [87].

Spinning disk confocal microscopy of fixed specimens and whole live worms was performed using a Nikon Eclipse Ti2 microscope equipped with a Plan Apo 60X 1.2 N.A. water immersion lens, a CSU-X1 confocal scanning unit (Yokogawa Electric Corporation, Tokyo Japan), and a Prime 95B CMOS camera (Teledyne Photometrics, Tucson, AZ). Excitation light was generated using 405 nm, 488nm, and 561 nm solid state lasers housed in an LU-NV laser unit. NIS-Elements software (Nikon Instruments, Inc, Tokyo, Japan) was used for image acquisition.

Time-lapse spinning disk confocal imaging of live specimens was performed on a Nikon TE2000U inverted microscope equipped with a Plan Apo 60X 1.4 N.A. oil immersion lens, a Thermo Plate heating/cooling stage (Tokai Hit, Japan), a CSU10 confocal scanning unit (Yokogawa Electric Corporation, Tokyo, Japan) and a C9100-13 EM-CCD camera. (Hamamatsu Photonics, Shizuoka, Japan). Excitation light was generated using 405 nm, 491 nm, and 561 nm solid state lasers controlled via a LMM5 laser launch (Spectral Applied Research, Ontario, Canada) and fed through a Borealis beam conditioning unit (Spectral Applied Research, Ontario, Canada). Images were acquired using MetaMorph software (Molecular Devices San Jose, CA).

Image processing was performed with either NIS-Elements software (Nikon Instruments, Inc, Tokyo, Japan) or Fiji.

## Immunoprecipitation-mass spectrometry

Whole worm extracts were prepared as described previously [88]. Eight milligrams of either *chd-1(bs122)* (control) or *chd-1(bs125)* (CHD-1::sfGFP) whole worm extracts and 80 microliters of GFP-Trap Magnetic Agarose beads (Chromotek) were used for each IP. The IPs were performed at 4˚C for 3 hours, washed twice with GFP-trap wash buffer (10 mM Tris-Cl pH 7.5, 150 mM NaCl, 0.5 mM EDTA), and suspended in 100 microliters of 1XTBS (50 mM Tris-Cl, pH 7.6, 150 mM NaCl). For mass spec, the samples were resuspended in a Tris/ Urea buffer, reduced, alkylated and digested with trypsin at 37˚C overnight. This solution was subjected to solid phase extraction to concentrate the peptides and remove unwanted reagents followed by injection onto a Waters NanoAcquity HPLC equipped with a self-packed Aeris 3 μm C18 analytical column 0.075 mm by 20 cm, (Phenomenex, Torrance, CA). Peptides were eluted using standard reverse-phase gradients. The effluent from the column was analyzed using a Thermo Orbitrap Elite mass spectrometer (nanospray configuration) operated in a data dependent manner for 120 minutes. The resulting fragmentation spectra were correlated against the known database using PEAKS Studio 8.5 (Bioinfomatic Solutions, Ontario, Canada) Scaffold Q+S (Proteome Software Inc., Portland, OR) was used to provide consensus reports for the identified proteins.

## Auxin treatment

To deplete CDK-2, we employed the auxin-inducible degradation system [71] and the *cdk-2 (kim31[cdk-2*::*aid*::*3xflag])* allele [89]. Worms were grown on standard MYOB plates until they reached the L3 stage, at which point they were transferred to MYOB plates supplemented with 1mM indole-3-acetic acid (Alfa Aesar, Haverhill, MA) and seeded with *E. coli* OP50. Worms were grown for two days at 20˚ C before being processed for quantitative immunoblotting.

## Quantitative immunoblotting

Samples for quantitative immunoblots were prepared from either whole worm lysates or worm extracts. For whole worm lysates, 100 gravid adults were washed twice with 1 ml of M9 buffer (22 mM $KH_2PO_4$, 22 mM $Na_2HPO_4$, 85 mM NaCl, 1 mM $MgSO_4$), resuspended in 40 μl of either NuPAGE LDS Sample Buffer (Invitrogen, Waltham, MA) or 4X Laemmli buffer (Bio-Rad Laboratories, Inc. Hercules, CA) and boiled at 95˚C for 10 minutes. The lysates were stored at -30˚C prior to gel electrophoresis. For worm extracts, worms were grown on eight 100 mm MYOB agar plates until the bacteria was exhausted. The starved L1 larvae were washed off the plates, transferred to 500 ml of liquid S-media containing *E.coli* NA22, and grown at 20˚C until they reached adulthood. Gravid worms were collected, and extracts prepared as described [88]. The concentrations of the extracts were determined using the Bio-Rad Protein assay kit (Bio-Rad Laboratories Inc. Hercules, CA). The extracts were then aliquoted into individual tubes, flash frozen in liquid nitrogen, and stored at -80˚C.

SDS-PAGE gel electrophoresis was performed using 12 to 16 μl of lysate or 50 μg of extract per lane. Samples prepared in NuPAGE LDS Sample Buffer were resolved on Novex 4–12% Bis-Tris precast gels (Invitrogen, Waltham, MA) and blotted to nitrocellulose using the iBlot semi-dry transfer system (Invitrogen, Waltham, MA) according to the manufacturer's instructions. Samples prepared in 4X Laemmli buffer were run on a 4–20% Mini-protean TGX precast gel (Bio-Rad Laboratories, Hercules, CA) and transferred to nitrocellulose using the Trans-Blot Turbo Transfer System (Bio-Rad Laboratories) according to the manufacturer's instructions. The membranes were blocked with Odyssey blocking buffer (LiCOR Biosciences, Lincoln, NE) and probed with a 1:1000 dilution of the following antibodies: anti-SPD-2 [41], anti-SAS-5 (an affinity purified rabbit antibody against the peptide n-

CPAERERRIREKYARRK-c produced by Yenzym Antibodies, LLC), anti-SAS-6 [29], anti-GFP (Roche, Indianapolis, IN), and anti-alpha-tubulin DM1A (Sigma-Aldrich, Inc). Anti-mouse 680, anti-guinea pig 800, and anti-rabbit 800 IRDye secondary antibodies (LiCOR Biosciences, Lincoln, NE) were used at a 1:14,000 dilution. Membranes were imaged using the Odyssey Clx imaging system (LiCOR Biosciences, Lincoln, NE). Quantitation of band intensities was performed using Fiji software [52] and normalized to the internal loading control.

## RNA-Seq

RNA samples (four biological replicates per genotype) were prepared from N2 and *chd-1 (bs122)* adult worms as described below for qRT-PCR. RNA quality was determined using an Agilent 2100 Bioanalyzer (Agilent, Santa Clara, CA). All RNA samples had an RNA integrity number (RIN) greater than or equal to 9.5. Libraries were constructed using the NEB Next Ultra II RNA library prep for illumina per the manufacturer's protocol (New England Biolabs Inc., Ipswitch, MA), and library concentrations measured using the 2100 Bioanalyzer. Libraries were pooled and sequenced on a HiSeq 2500 Sequencing System (Illumina, San Diego, CA), generating an average of 22 million reads for the N2 library (range 17–28 M reads) and an average of 23 million reads for the *chd-1(bs122)* library (range 21–26 M reads). FASTQ files were aligned to ce10 genome using BBMAP, Version 36.02. Gene count table, differential gene expression analysis and PCA analysis were performed using the Genomatix Genome Analyzer (GGA) [https://www.genomatix.de/solutions/genomatix-genome-analyzer.html].

## qRT-PCR

Approximately 300 day-one adult (72 hours after plating as L1) animals were collected for RNA extraction by washing three times with M9 prior to suspension in 1mL Trizol Reagent (Thermo-Fisher Scientific #15596026). Nucleic acids were isolated via chloroform extraction and RNA was isolated with the Qiagen RNEasy Mini (Qiagen #74104) including on-column DNase I digestion. RNA was assessed for purity and concentration with a NanoDrop Lite. cDNA synthesis was performed with the iScript cDNA Synthesis Kit (Bio-Rad #1708891). qRT-PCR was performed with SYBR Select Master Mix (ThermoFisher #4472908) on a Bio-Rad CFX384 Touch Real-Time PCR Detection System (Bio-Rad # 1855485). Six biological replicates of each genotype were run in triplicate for each gene. Fold change was determined by the ΔΔct method with Δct compared to *act-1*. A list of primer sequences used for qRT-RCR can be found in **S5 Table**.

## Protein structure prediction and alignment

The protein structure of *C. elegans* CHD-1 was predicted using SWISS-MODEL [90–92]. Alignment of the predicted *C. elegans and* known yeast CHD-1 (PDB ID: 3MWY) [57] structures was performed using PyMOL software (The PyMOL Molecular Graphics System, Version 1.2r3pre, Schrödinger, LLC.).

## Statistics and scatter plots

Statistical analysis and the generation of scatter plots were performed using Prism 8.3 (GraphPad Software, San Diego, CA). Differences in gene expression were assessed as parallel t-tests with a Holm-Sidak correction for multiple comparisons using the Graphpad Prism 8 platform.

## Supporting information

**S1 Fig. Loss of CHD-1 suppresses the spindle-assembly defect of *zyg-1(it25)* and confers a cold-sensitive fecundity defect.** (A) Frames from 4D DIC imaging of a *zyg-1(it25)* embryo

(top) and a *chd-1(ok2798); zyg-1(it25)* double mutant embryo (bottom). The first time point shows each embryo early at the two-cell stage. The second time point shows embryos in second mitosis with spindle poles in AB cell indicated by arrowheads. The third time point shows embryos after exiting second mitosis. The *zyg-1(it25)* embryo assembles monopolar spindles and fails to exit the two-cell stage while the *chd-1(ok2798); zyg-1(it25)* embryo assembles bipolar spindles and progress to the four-cell stage. Time is in min:sec and is relative to AB nuclear envelope breakdown (t = 0). Scale bar, 10 μm. (B) Embryonic viability among the offspring of *chd-1(bs122)* hermaphrodites measured at the indicated temperatures. Each data point represents the percent embryonic viability among the progeny of an individual hermaphrodite over the course of its reproductive lifespan. Bars indicate mean and standard deviation. (C) The brood sizes of *chd-1(bs122)* hermaphrodites were measured at the indicated temperature. Each dot represents the brood size of a single hermaphrodite. Bars indicate mean and standard deviation. *** p<0.0001, *p,0.05, ns = not significant, two-tailed t test.
(TIF)

**S2 Fig. Mass spectrometry identification of CHD-1::sfGFP interacting proteins (A) Proteins were immunoprecipitated from whole worm extracts with an anti-GFP antibody.** Strain OC829 (*chd-1(bs122)*) expresses sfGFP from the endogenous *chd-1* locus and served as a negative control. Strain OC798 (*chd-1(bs125)*) expresses CHD-1::sfGFP from the endogenous locus. The blot was probed for GFP to validate the pull down. (B) Proteins identified as significantly enriched in the CHD-1::sfGFP immunoprecipitate.
(TIF)

**S3 Fig. Spinning disk confocal images of worms expressing wild-type CHD-1::GFP (chd-1 (bs125)) or a helicase-dead version CHD-1(D538N)::GFP (chd-1(bs123)).** Both proteins localize similarly. Shown are nuclei of the distal germ line (p) and for CHD-1(D538N)::GFP, oocyte nuclei (o). The wild-type transgene is also expressed in oocyte nuclei but these nuclei are not present in the optical plane shown. Scale bar = 25μm.
(TIF)

**S4 Fig. Loss of CHD-1 results in a modest increase of centrosome-associated ZYG-1 levels and lower total levels of SAS-5. (A)** Representative images of embryos immunostained for ZYG-1::SPOT (red), SAS-4 (green) and DNA (blue). Each image is a maximum intensity projection of 10 focal planes. Top left, a *zyg-1(bs197[zyg-1::spot])* embryo and bottom left, a *chd-1 (bs122); zyg-1(bs197[zyg-1::spot])* embryo. Insets are three-fold magnified images of centrioles stained with ZYG-1(Z1) or SAS-4 (S4). Quantification of the raw integrated intensity (expressed in arbitrary units or a.u.) of ZYG-1::SPOT staining at centrioles. Each dot represents a single centriole in anaphase. Bar = 10 μm. **p = 0.007, unpaired t test (B). Left, quantitative immunoblot of SAS-5 showing isoforms a and b. Each lane is an independent sample of 100 gravid adults. Right, Quantitation of SAS-5a showing relative levels in wild-type and *chd-1 (bs122)* animals.
(TIF)

**S5 Fig. The effect of the chd-1(bs122) mutation on the centrosome-associated levels of SPOT::SAS-6, SAS-4, and GFP::SPD-2. (A)** Left. Representative images of embryos immunostained for SPOT::SAS-6 (red), SAS-4 (green) and DNA (blue). Each image is a maximum intensity projection of 25 focal planes. Top, a *sas-6(bs188 [spot::SAS-6])* embryo and bottom, a *chd-1(bs122); sas-6(bs188 [spot::SAS-6])* embryo. Insets are 1.5-fold magnified images of centrioles stained for SPOT::SAS-6 (S6) or SAS-4 (S4). Scale bar, 10 μm. Right. Quantification of the raw integrated intensity (expressed in arbitrary units or a.u.) of SPOT::SAS-6 and SAS-4 at centrioles. Each dot represents a single centriole in anaphase. Not significant (ns) as

determined by an unpaired t test. (B) Quantitative fluorescence microscopy of a *gfp::spd-2* transgene in *chd-1(bs122)* heterozygous (control) and homozygous embryos. As shown in the schematic, the transgene is expressed under control of the *pie-1* promoter and 3' utr and its expression is negatively affected by loss of CHD-1. Representative images (left, scale bar = 10 μm) and quantitation (right) are shown. Each dot represents a single centrosome. Bars indicate mean and standard deviation. ***p<0.0001, unpaired t test with Welch's correction. (C) Quantitative immunoblot of endogenous SPD-2 and GFP::SPD-2 levels in *chd-1(bs122)* heterozygotes and homozygotes. Note that endogenous SPD-2 is elevated almost two-fold in the *chd-1(bs122)* homozygotes relative to heterozygous siblings, while the level of GFP::SPD-2 is reduced.
(TIF)

**S6 Fig. Extra centrioles in a chd-1; dpl-1 double mutant arise in association with pre-existing centrioles.** Select time points from a recording of a *chd-1(bs122); dpl-1(bs169)* embryo expressing a *spd-2::mCherry* transgene that was grown and recorded at 12.5˚ C. The first time point is at the onset of the two-cell stage. Initially, two SPD-2::mCherry-positive bodies are visible in the anterior blastomere (arrowheads, 00:00 and 05:00). Shortly thereafter, a third SPD-2::mCherry-positive body becomes visible (arrowheads, 08:00 and 16:00). These structures mature into centrosomes that form a tripolar spindle (23:00). Subsequently, each centrosome resolves into two structures (43:00 and 50:00), indicating that all three centrosomes are capable of duplication. The two centrosomes of the posterior blastomere are visible from timepoints 08:00 to 23:00. Each image is a maximum intensity projection of 6 focal planes. Time is in minutes: seconds and is relative to first frame. Scale bar = 10 μm. Right.
(TIF)

**S1 Table. *chd-1* Downregulated Genes.**
(XLSX)

**S2 Table. *chd-1* Upregulated Genes.**
(XLSX)

**S3 Table. *C. elegans* Strains.**
(DOCX)

**S4 Table. CRISPR Reagents.**
(DOCX)

**S5 Table. qRT-PCR Primers.**
(DOCX)

**S6 Table. Numerical data.**
(XLSX)

## Acknowledgments

We would like to thank Tony Hyman for generously providing the anti-GIP antibody, Yumi Kim for providing the *cdk-2::aid::3xflag* strain, Jennifer Patterson-West for assisting with analysis of protein structure, and Harold Smith and Sijung Yun for assistance with RNA-Seq. Some strains were provided by the CGC, which is funded by the NIH Office of Research Infrastructure Programs (P40 OD010440), and by the National BioResource Project (Japan).

## Author Contributions

**Conceptualization:** Jyoti Iyer, Bruce Bowerman, Kevin F. O'Connell.

**Data curation:** Sarah Guagliardo, Kevin F. O'Connell.

**Formal analysis:** Jyoti Iyer, Lindsey K. Gentry, Sarah Guagliardo, Peter A. Kropp, Prabhu Sankaralingam, Eric Spooner, Kevin F. O'Connell.

**Funding acquisition:** Kevin F. O'Connell.

**Investigation:** Jyoti Iyer, Lindsey K. Gentry, Mary Bergwell, Amy Smith, Sarah Guagliardo, Peter A. Kropp, Prabhu Sankaralingam, Eric Spooner, Kevin F. O'Connell.

**Methodology:** Jyoti Iyer, Yan Liu, Eric Spooner.

**Project administration:** Kevin F. O'Connell.

**Resources:** Yan Liu, Kevin F. O'Connell.

**Supervision:** Jyoti Iyer, Kevin F. O'Connell.

**Validation:** Jyoti Iyer, Kevin F. O'Connell.

**Visualization:** Kevin F. O'Connell.

**Writing – original draft:** Jyoti Iyer, Kevin F. O'Connell.

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
