## [Decision Letter · Decision Letter 0]

1 Oct 2021

Dear Dr O'Connell,

Thank you very much for submitting your Research Article entitled 'The Chromatin Remodeling Protein CHD-1 and the EFL-1/DPL-1 Transcription Factor Cooperatively Down Regulate CDK-2 to Control SAS-6 Levels and Centriole Number' to PLOS Genetics.

The manuscript was fully evaluated at the editorial level and by three independent peer reviewers. The reviewers appreciated the attention to an important problem, but raised some substantial concerns about the current manuscript. In particular, Reviewer 2 raises some important points to address regarding whether the increase in SAS-6 and SPD-2 levels is the cause of the observed phenotypes. Based on the reviews, we will not be able to accept this version of the manuscript, but we would be willing to review a much-revised version. We cannot, of course, promise publication at that time.

If you decide to revise the manuscript for further consideration at PLOS Genetics, please aim to resubmit within the next 60 days, unless it will take extra time to address the concerns of the reviewers, in which case we would appreciate an expected resubmission date by email to plosgenetics@plos.org.

[LINK]

We are sorry that we cannot be more positive about your manuscript at this stage. Please do not hesitate to contact us if you have any concerns or questions.

Yours sincerely,

Jeremy Nance

Associate Editor

PLOS Genetics

Gregory P. Copenhaver

Editor-in-Chief

PLOS Genetics

Reviewer's Responses to Questions

**Comments to the Authors:**

Reviewer #1: In the study titled “The Chromatin Remodeling Protein CHD-1 and the EFL-1/DPL-1 Transcription Factor Cooperatively Down Regulate CDK-2 to Control SAS-6 Levels and Centriole Number”, Iyer et al. identified that SAS-6 levels is regulated through CHD-1 and EFL-1/DPL-1 in C.elegans. The title concludes that centrosome duplication is involved, for the authors to directly conclude this it would be useful to demonstrate over duplication of centrioles using live cell imaging versus a cytokinesis defect. Some of the example images included demonstrate cells with clear supernumerary centrosomes (Fig 5D). If the panel was labeled clearly as a 2-cell stage embryo it could demonstrate that the phenotype was due to over-duplication. However, it would strengthen the study and support the main conclusion in the title to demonstrate overduplication through live-cell imaging. It was also intriguing that previous studies found that CHD-1 can have an interaction with the centrosome and that this interaction can potentially confer function. The authors herein do not find any interactions of CHD-1 with centrosome proteins through IP analysis. It would be useful for the authors to expand upon these ideas more. For instance, is there differences between the C. elegans centrosome protein composition and Xenopus centrosome that could account for this? Are there experiments that the authors could perform to distinguish if CHD-1 has a role at the centrosome separate from its role as a chromatin remodeling protein? Overall, the paper was well written and with expanded discussion and live-cell imaging studies demonstrating over-duplication, it is acceptable for publication at PLOS Genetics.

Specific Comments:

• Line 190-192, it is claimed that absence of CHD-1 does not cause death or sterility on an organism level, but does it cause defects in mitotic progression on the cellular level that could be discussed?

• Lines 270-271, its stated that CHD-1 is a low abundance protein based on a band against GFP. Can endogenous untagged CHD-1 be examined?

• Model and immuno-fluorescent resolution were low and it was difficult to visualize images. Labeling for these images can be improved upon (e.g. time stamps, what colors mean in images), and separating channels for clear visualization of centrosome components being perturbed or numbers altered (most notably in Figures 1E, 5D).

• Lines 347-353 (Figure 3A and 3B), expression of SAS-6 and SPD-2 were analyzed. Are other centrosome proteins affected or is CHD-1-loss specific to just these two?

• Figure 1 E(F), figure description was mislabeled, description of Figure 1F should be for figure 1E.

• Figure 5D needs to be better labeled and adding an image of wild type negative control as comparison. Figure legends for Figure 5D was mislabeled as 5E. Different labeling of figure 5D could improve ease of reading. For example: dpl-1 can be changed to dpl-1(bs169)).

• Model 6E was difficult to read, consider revising.

Reviewer #2: Central to cell replication is the duplication of microtubule-based structure called centrioles that lie at the heart of the spindle poles. Although enigmatic, the process of centriole duplication has been genetically dissected over the last two decades, including by work from the O’Connell lab. Here, Iyer et al. investigate the regulation of the expression of the building blocks of the centrioles, uncovering a role for the chromatin remodeling protein CHD-1, the transcription factor EFL-1/DPL-1, and the kinase CDK-2. Mutations in chd-1 suppress lethality associated with the loss of the kinase zyg-1/Plk4 and restore some “normal” centriole duplication. CHD-1 depletion also leads to increased protein levels, by not transcription, of SPD-2 and SAS-6 and acts in parallel to EFL-1/DPL-1 to downregulate the transcription of CDK-2. Loss of CDK-2 leads to the downregulation of SAS-6 protein levels. Thus, the authors propose that CHD-1 and EFL-1/DPL-1 function in parallel to regulate the expression of CDK-2, which in turn promotes SAS_6 expression to directly regulate centriole duplication. The experiments appear well executed and controlled and the topic area provides a conceptual advance. I think the paper would be suitable for publication but recommend a few changes:

- The authors clearly demonstrate that SAS-6 and SPD-2 levels are upregulated in the various depletion cases (cdk-2, chd-1). However, the connection between the upregulation of these protein levels and the ensuing phenotypes is tenuous. Can the author test whether partial depletion of spd-2 or sas-6 (either the spd-2 ts mutant or RNAi) suppresses the cdk-2 or chd-1 phenotypes.

- Similarly, the connection between the centriole duplication-based phenotypes and cdk-2 could be strengthened. Does overexpression of cdk-2 on its own produce a centriole overduplication phenotype? Are the effects of cdk-2 specific? For example, loss of CDK-2 following auxin treatment might just lead to cell cycle arrest that would in turn decrease SAS-6 levels. Are other proteins (maybe SAS-5) unaffected in this depletion case? Do the authors see the same enhancement of the chd-1/zyg-1 or dpl-1/zyg-1 phenotype following loss of CDK-1 for example?

- Figure 1E: needs a better description of duplication process to make these images understandable to non-C. elegans people. The images need time stamps and indication that the centrosomes and not centrioles are shown and the relation of the results here to centriole duplication defects.

- Although likely chronological, I find the points of Figure 3D and E detracting from the main thread of the story. Figure 3F supports the main conclusion and Figure 3D and E show that endogenous flanking sequence is important. I would switch the order of the presentation of this information and perhaps move 3D and E to the supplement.

- Line 479: suggesting that overexpression of SAS-6 is toxic—this is a leap based on the information presented here as I am sure many transcripts are affected following the loss of both CHD-1 and DPL-1

- Can you quantify the effects seen in Figure 5D: in the one example of dpl-1 single mutant, supernumerary centrosomes appear to coalesce at the poles while the chd-1;dpl-1 double appear multipolar. Is this the case?

Minor points:

- There are several instances where data is referenced but not shown:

1.Where is the data for lines 205-210?

2.Where are the IP data, i.e. the gel? How can we assess the legitimacy of the results of this experiment without being able to assess if the IP worked.

3. Where is the data to support the diagnosis of successful cytokinesis (line 493-496)

- Figures are low resolution. The images are particularly hard to interpret at this resolution. This may just be the reviewer copy, but I would make sure that the final veriosn of the figures are at higher resolution. I would also encourage the authors to include larger and higher resolution images of their photographic data throughout. There is a lot of wasted white space in the figures that could be used for making images larger without having to make the overall figures bigger.

- Figure S3A - SAS-4 also seems upregulated. Is this the case?

- Line 90: in a mice

- Figure 1D: This should be represented as % events rather than number of events

- Line 357 needs a comma

- Line 488: please define GIP-1 ortholog, i.e. g-TuRC component GCP3

Reviewer #3: This is a very interesting study that explores the regulation of centriole number maintenance by transcriptional control pathways . In this manuscript Iyer and colleagues identify two pathways working in parallel and converging on the kinase CDK-2, which in turn regulates SAS-6 protein levels to prevent centriole overduplication in C. elegans. The authors identify CHD-1 as one of the regulatory pathways by a suppression phenotype of the zyg-1(it25) mutation and devote substantial effort to analyze its role in the regulation of centriole core components. Further, they demonstrate that the E2F-DPL-1 complex acts in parallel to CHD-1 in the same process. This study is in line with previous works demonstrating a role of CDK-2 in centriole assembly in vertebrates, but it adds a mechanistic view to the regulation. I think the paper is of high interest. Nevertheless, the authors should address the following concerns before the manuscript can be considered suitable for publication in PLOS Genetics.

Major comments:

1. On page 16, the authors engineer a mutant to test the requirement of the ATPase activity of the CHD-1 helicase and come to the conclusion that it is required for suppression of the zyg-1(it25) lethality. On page 29 authors mention that in other systems CHD1 can have functions which are not associated with the chromatin regulation. I wonder if the authors tried to disrupt the DNA-binding domain in CHD-1 and to test whether this would have a comparable effect as the mutation in the ATPase domain. Such an experiment could help to discriminate whether the regulation of centriole biogenesis is indeed through chromatin.

2. On pages 16-17, the authors find that in chd-1(bs122) mutants protein levels of SAS-6 and SPD-2 are upregulated. The authors then explore in detail the regulation of SPD-2 levels at the centrosome using different constructs and come to the conclusion that SPD-2 is upregulated through its promoter and 3’UTR. These experiments are done in the presence of the endogenous protein, which complicates the analysis. I wonder why the authors did not use an available endogenously CRISPR-tagged protein or immunostainings to measure endogenous SPD-2 levels at the centrosome. Alternatively one could down-regulate endogenous SPD-2 by RNAi given that the SPD-2:mCherry transgene is RNAi-resistant.

3. It is puzzling that the elevated SAS-6 protein level in chd-1(bs122) does not translate into higher levels of SAS-6 at the centrioles. This could be due to the fact that only a finite number of SAS-6 molecules can be incorporated in a given centriole or alternatively the decreased SAS-5 protein level could be a limiting factor. The authors should at least discuss how the decreased SAS-5 levels fit into the model of elevated SAS-6 levels in chd-1(bs122) single mutants.

4. While the findings of elevated SAS-6 overall levels in the chd-1(bs122) mutants is an important result, the authors should test for SAS-5 and SAS-6 levels at the centrioles in the sensitized chd-1(bs122) zyg-1(it25) double mutant background, where an increase might be more evident.

5. The authors demonstrate that chd-1(bs122) and dpl-1(RNAi) in an otherwise wild type background (non-sensitized) have a synergistic effect on embryonic viability and that chd-1(bs122) enhances the lethality of dpl-1(bs169). This strongly indicates that in the double mutants SAS-6 protein levels are even more elevated and reach a toxic level. If this is the underlying cause of centriole overduplication, authors should demonstrate elevated SAS-6 protein levels in this combined genetic background.

6. The authors claim to see supernumerary centrosomes in the chd-1;dpl-1 double mutants. While this is a very important and exciting finding, some more evidence is needed. Extra gamma tubulin foci can be a consequence of different processes like fragmentation of the PCM, fragmentation of centrioles, premature disengagement of both or one centriole pair or failed cytokinesis. The authors argue that the extra gamma-tubulin foci are only detectable from the second division on and that the absence of binucleated cells indicates a successful cytokinesis. It is difficult to draw a conclusion using only gamma tubulin as a marker and fixed immunostaining samples. Especially the image of the dpl-1 single mutant (Fig. 5D top right) shows a small and big gamma tubulin focus, which resembles a prematurely disengaged centriole pair. To conclude that the extra foci are for sure a consequence of centriole overduplication, authors need to provide some more evidence. They could use additional centriole markers and/or perform live cell imaging to precisely determine when and where these extra foci/centrioles appear.

7. In the discussion no explanation is offered for some of the rather surprising and puzzling results. For instance, how do the authors reconcile the decreased SAS-5 levels in chd-1(bs122) with the model of centriole overduplication and how do the authors explain the discrepancies in the RNAseq and the qRT-PCR data?

Minor issues:

1. Figure 1 the labels of D, E and F need to be adjusted according to the figure legends. Figure 1E would be better represented in percentages, as a stacked column. In Figure 1F it would be helpful if the authors add the time stamps to the stills.

2. In Figure 2B the insets are hardly visible, they can be increased in size.

3. Line 324/Figure S2 Authors state that CHD-1(D538N)::sfGFP localization is not altered. In the images it seems that the mutant version localizes much stronger to the nuclei of the oocytes than the wild type protein. Authors should comment on this in the text.

4. In Table 1 an indication of the present (%) of for a given phenotype would be helpful.

5. For figure S3 it should be specified in the methods which SAS-5 antibody was used and which isoform was quantified to measure SAS-5 levels.

6. In Figure 5D image of the embryos can be increased for better visibility.

7. For completion the authors could include an arrow for the non-CDK-2 input of CHD-1 in the model.

**Have all data underlying the figures and results presented in the manuscript been provided?**

Reviewer #1: Yes

Reviewer #2: Yes

Reviewer #3: Yes

PLOS authors have the option to publish the peer review history of their article (what does this mean?). If published, this will include your full peer review and any attached files.

Reviewer #1: No

Reviewer #2: No

Reviewer #3: No

---

## [Decision Letter · Decision Letter 1]

17 Mar 2022

Dear Dr O'Connell,

We are pleased to inform you that your manuscript entitled "The Chromatin Remodeling Protein CHD-1 and the EFL-1/DPL-1 Transcription Factor Cooperatively Down Regulate CDK-2 to Control SAS-6 Levels and Centriole Number" has been editorially accepted for publication in PLOS Genetics. Congratulations!

Yours sincerely,

Jeremy Nance

Associate Editor

PLOS Genetics

Gregory P. Copenhaver

Editor-in-Chief

PLOS Genetics

Comments from the reviewers (if applicable):

Reviewer's Responses to Questions

**Comments to the Authors:**

Reviewer #1: The authors addressed all my concerns thoughtfully in their rebuttal and is an exciting take on regulatory mechanisms of centriole duplication. I found the updated figures were beautiful and well organized.

Reviewer #2: Here, Iyer et al. submit a revised version of their manuscript. Their original manuscript had compelling data and I find the revised manuscript clear and the experiments well executed and controlled. I appreciate the technical challenges associated with my suggestions, but look forward to future experiments modulating the levels of SAS-6 to determine whether an increase in SAS-6 levels is sufficient to produce centriole overduplication on its own or whether low level depletion of SAS-6 can suppress the overduplication phenotypes seen in this paper. I recommend publishing the current manuscript in PLOS Genetics.

Reviewer #3: The authors performed a series of additional experiments which have significantly improved the study. The authors addressed all my comments and the paper is ready for publishing.

**Have all data underlying the figures and results presented in the manuscript been provided?**

Reviewer #1: Yes

Reviewer #2: Yes

Reviewer #3: Yes

PLOS authors have the option to publish the peer review history of their article (what does this mean?). If published, this will include your full peer review and any attached files.

Reviewer #1: No

Reviewer #2: No

Reviewer #3: No

**Data Deposition**

http://datadryad.org/submit?journalID=pgenetics&manu=PGENETICS-D-21-01179R1

**Press Queries**

---

## [Editor Report · Acceptance letter]

31 Mar 2022

PGENETICS-D-21-01179R1 

The Chromatin Remodeling Protein CHD-1 and the EFL-1/DPL-1 Transcription Factor Cooperatively Down Regulate CDK-2 to Control SAS-6 Levels and Centriole Number 

Dear Dr O'Connell, 

We are pleased to inform you that your manuscript entitled "The Chromatin Remodeling Protein CHD-1 and the EFL-1/DPL-1 Transcription Factor Cooperatively Down Regulate CDK-2 to Control SAS-6 Levels and Centriole Number" has been formally accepted for publication in PLOS Genetics! Your manuscript is now with our production department and you will be notified of the publication date in due course.

With kind regards,

Kata Acsay

PLOS Genetics

On behalf of:
